# Spatial transcriptomics reveals niche-specific enrichment and vulnerabilities of radial glial stem-like cells in malignant gliomas

Yanming Ren [1,6], Zongyao Huang [1,6], Lingling Zhou [1,6], Peng Xiao [1,6], Junwei Song [2,6], Ping He[2], Chuanxing Xie[1], Ran Zhou[1], Menghan Li[1], Xiangqun Dong[1], Qing Mao [3], Chao You[3], Jianguo Xu[3], Yanhui Liu[3], Zhigang Lan[3], Tiejun Zhang[3], Qi Gan[3], Yuan Yang[3], Tengyun Chen[3], Bowen Huang[3], Xiang Yang[3], Anqi Xiao[3], Yun Ou[4], Zhengzheng Su[4], Lu Chen [2] ✉, Yan Zhang [5] ✉, Yan Ju [3] ✉, Yuekang Zhang [3] ✉ & Yuan Wang [1] ✉

Diffuse midline glioma-H3K27M mutant (DMG) and glioblastoma (GBM) are the most lethal brain tumors that primarily occur in pediatric and adult patients, respectively. Both tumors exhibit significant heterogeneity, shaped by distinct genetic/epigenetic drivers, transcriptional programs including RNA splicing, and microenvironmental cues in glioma niches. However, the spatial organization of cellular states and niche-specific regulatory programs remain to be investigated. Here, we perform a spatial profiling of DMG and GBM combining short- and long-read spatial transcriptomics, and single-cell transcriptomic datasets. We identify clinically relevant transcriptional programs, RNA isoform diversity, and multi-cellular ecosystems across different glioma niches. We find that while the tumor core enriches for oligodendrocyte precursor-like cells, radial glial stem-like (RG-like) cells are enriched in the neuron-rich invasive niche in both DMG and GBM. Further, we identify niche-specific regulatory programs for RG-like cells, and functionally confirm that FAM20C mediates invasive growth of RG-like cells in a neuron-rich microenvironment in a human neural stem cell derived orthotopic DMG model. Together, our results provide a blueprint for understanding the spatial architecture and niche-specific vulnerabilities of DMG and GBM.

Malignant gliomas, including diffuse midline glioma-H3K27M mutant (DMG) and glioblastoma (GBM), are the most frequent and lethal primary brain tumors[1]. DMG is a new entity added to malignant gliomas in the recent WHO classification, which includes the tumor type formerly known as DIPG[1]. It most commonly occurs in children, typically in midline brain regions including the brain stem, thalamus, and cerebellum[1,2]. It harbors a signature lysine 27-to-methionine mutation in histone H3 (*H3K27M*), often accompanied by *TP53* mutations and

[1]Department of Neurosurgery, State Key Laboratory of Biotherapy and Cancer Center, West China Hospital, Sichuan University, Chengdu 610041, China. [2]Key Laboratory of Birth Defects and Related Diseases of Women and Children of MOE, West China Second Hospital, Sichuan University, Chengdu 610041, China. [3]Department of Neurosurgery, West China Hospital, Sichuan University, Chengdu, Sichuan 610041, China. [4]Department of Pathology, West China Hospital, Sichuan University, Chengdu, Sichuan 610041, China. [5]National Clinical Research Center for Geriatrics, West China Hospital, Sichuan University, Chengdu, Sichuan 610041, China. [6]These authors contributed equally: Yanming Ren, Zongyao Huang, Lingling Zhou, Peng Xiao, Junwei Song. ✉e-mail: luchen@scu.edu.cn; yanzhang@scu.edu.cn; juyanwestchina@126.com; 2012zykyx@sina.cn; wangyuan@scu.edu.cn

*PDGFRA* amplification[3]. GBM, on the other hand, primarily occurs in the frontal and temporal lobes of adult brains, and is driven by more complex combinations of genetic events[1,4,5]. Despite the differences in age of onset, anatomic location, and genetics, both tumors have very poor prognosis, with a median overall survival of 9–14 months under the standard treatment paradigm of surgery and chemo/radiation therapy[1,2,4]. The main challenges underlying the treatment failure for these tumors include high degree of heterogeneity and their characteristic infiltrative growth[1,2,4].

The heterogeneity of DMG and GBM is mainly shaped by distinct genetic/epigenetic drivers, transcriptional programs including RNA splicing, and microenvironmental cues. A series of single-cell RNA-sequencing (scRNA-seq) studies have elegantly revealed diverse cellular states and cell types within DMG and GBM, which exhibit similarities to normal cells in developing and adult brains such as astrocytes (ACs), oligodendrocytes (OCs), and oligodendrocyte precursor cells (OPCs)[6–11]. In GBM, Bhaduri et al. identified a subpopulation of cancer stem cells resembling embryonic neural stem cells outer radial glia (RG), which is implicated in glioma progression and invasion[8]. However, it is unknown whether a similar cell population exists in DMG considering DMG is mainly comprised of proliferating OPCs based on single-cell analysis[6]. More importantly, the spatial organization of cellular states and cell types in DMG and GBM remains largely elusive. It has been well-established that glioma cells reside in spatially segregated regions called niches, forming complex ecosystems with microenvironmental cells including neurons, glia, blood vessel, and immune cells to support their growth and treatment resistance[12–15]. Thus, it is important to characterize the ecosystems within these niches in a spatial setting, which is lacking in scRNA-seq analyses.

The advancement of spatial transcriptomics provides a high-throughput method to interrogate tumor heterogeneity in a spatial context[16–19]. This technique, when coupled with the newly developed long-read sequencing, can simultaneously reveal the spatial heterogeneity of gene expression as well as differential transcript isoform expression resulting from alternative RNA splicing (AS)[20–22]. Compared to short-read sequencing, long-read sequencing has the unique advantage to discover rare isoforms, accurately quantify isoform expression, and detect differential AS events, since it requires fragmentation and reassembly of genomic sequences in short-read sequencing[23].

In this study, we perform a spatial profiling of DMG and GBM through short- and long-read spatial transcriptomics to decode the ecosystems and identify specific regulatory programs in distinct glioma niches.

## Results

We collected five DMG, three IDH-wildtype primary GBM (GBM$^{IDHwt}$), two IDH-mutant secondary GBM (GBM$^{IDHmut}$), and one peritumor samples from ten patients requiring surgical resection (Fig. 1a, Supplementary Figs. 1a and 2a, and Supplementary Data 1). Among these tumors, 9/10 are *TP53*-mutant. *H3K27M* mutation is exclusively found in DMG samples (5/5). To establish a dataset for spatially resolved gene and isoform expression, we performed short-read and long-read spatial transcriptomic sequencing (ST-seq) on the same tissue sections using the 10X Visium platform (Fig. 1a, Supplementary Data 2).

### Sample-wise spatially informed clustering and identification of malignant spots

Short-read ST-seq yields 26,460 high-quality transcriptomes/spots ("Methods"). In the current experimental setting, mRNAs may bleed between and among nearby spots (spot swapping) causing substantial contamination[24]. To correct for spot swapping, we used a recently published tool, SpotClean[24], to generate adjusted gene expression matrices for subsequent analyses (Fig. 1a). For example, the expression pattern of *OLIG2*, a oligodendrocyte lineage and glioma marker[25],

became more specific in the tumor core and the cerebellar white matter in DMG1 after SpotClean (Fig. 1a). Spots from different samples are horizontally integrated in the transcriptional space by Harmony[26] (Fig. 1b). To integrate both transcriptional space and Cartesian space for spatially informed spot clustering, we tested several recently developed spatially aware tools such as Seurat[27], BayesSpace[28], SpatialPCA[29], Spruce[30], SpatialDE2[31], and BANKSY[32] ("Methods"). Since the DMG1 sample contains a significant portion of normal cerebellum tissue with clearly demarcated anatomic domains, we used DMG1 as a benchmark to compare the clustering results, and found that the clusters generated by Banksy best correlate with anatomical domains in DMG1. Thus, we performed BANKSY on spots from each sample, generating unique spatial clusters that can be mapped onto distinct histopathological regions (Supplementary Figs. 1b, 2b, and "Methods").

To identify malignant spots with relatively high tumor cell content, we performed inferCNV analysis using histologically normal peritumor tissue as a reference[11]. We identified spots with broad chromosome number variations (CNVs) characteristic of malignant cells, including hallmark Chr 1q, 2 gain, and Chr 5q loss in DMG, as well as Chr 7 gain and Chr 10 loss in GBM[3,5] (Supplementary Figs. 1b and 2b). In each sample, we designated prominent CNV events shared among spots containing tumor cells based on histopathology, such as Chr7 amplification in GBM, as tumor signature CNV events (Fig. 1c). We then estimated the tumor content in each spot based on their score of the tumor signature CNV events (Fig. 1c, d, and "Methods"). Of note, the peritumor sample GBM5_2 also contained an area of spots with CNVs (Fig. 1b, d, and Supplementary Fig. 2b), which were excluded from the normal reference ("Methods"). In addition, although 10X Visium platform is based on 3′ sequencing, we can infrequently obtain reads at the K27 locus of *H3F3A* (-1 per 100,000 reads), and identify A-to-T point mutation (*H3K27M*) in spots from DMG but not GBM samples (Supplementary Fig. 1c). The majority of spots with *H3K27M* mutation also exhibit broad CNVs (Fig. 1e), independently confirming the accuracy of our CNV call. However, the low detection rate makes *H3K27M* mutation less reliable to estimate the tumor content. Thus, we used CNV-based tumor content estimation to filter out malignant spots for subsequent analyses ("Methods").

### Glioma niche-specific transcriptional programs

To determine spatial transcriptional programs, we first analyzed patient samples individually by BANKSY to group malignant spots into spatially informed clusters (Fig. 1a), and identified marker gene sets (cluster signature) for each cluster except those with fewer than 50 significant genes ("Methods"). This resulted in 48 cluster signatures from 10 tumor samples. To identify recurrent spatial transcriptional programs across samples, we horizontally integrated these cluster signatures within the transcriptional and Cartesian space ("Methods"). In the transcriptional space, we identified four meta-modules, which are present in all glioma samples (Fig. 2a and Supplementary Data 3). In the Cartesian space, we used two methods to perform spatially weighted correlation and hierarchical clustering to confirm the spatial segregation of these four modules (Fig. 2b and "Methods"). To test whether these modules are conserved across different glioma types, we quantified the contribution of cluster signatures by tumor type, and found that all four modules consist of clusters from three glioma types (Fig. 2c). We further performed horizontal integration by tumor types, and found tumor type-specific modules well correlate with pan-glioma modules (Supplementary Fig. 3a, b). To investigate the biological significance of these modules, we first performed Gene Ontology enrichment analysis (Supplementary Data 4). Module 1 enriches genes associated with normal gliogenesis and gliomagenesis (e.g. *ALDOC, OLIG2, PDGFRA, ASCL1, and SOX10*). Interestingly, most cell cycle-related genes (e.g. *CDK1, TOP2A*) are only found in Module 1. Module 2 enriches genes related to vasculogenesis and endothelial cells (e.g.

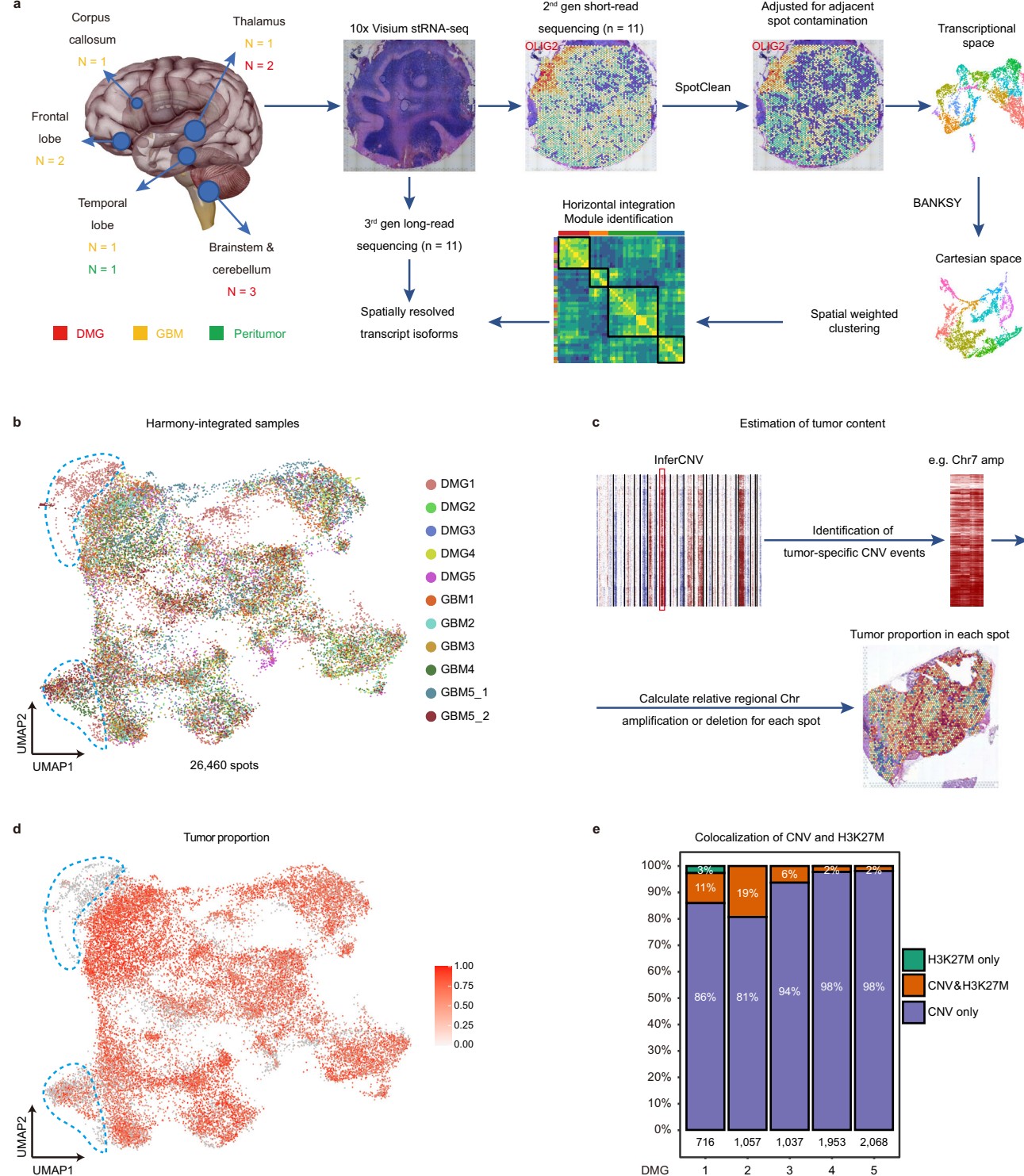

**Fig. 1 | Overview of workflow and sample information. a** Illustration of the overall workflow. Examples images for SpotClean-adjusted gene expression (*OLIG2*) and BANKSY-based spatial clustering for DMG1 are shown. **b** UMAP of all Harmony-integrated spatial transcriptomic spots (*n* = 26, 460) from all samples (*n* = 11). Colors represent different samples. Clusters of peritumor spots are highlighted by blue dashed lines. **c** Overview of the workflow for CNV-based estimation of tumor cell content. GBM4 is shown as an example. Colors indicate the predicted tumor cell content score. **d** UMAP of all Harmony-integrated spatial transcriptomic spots from all samples (*n* = 26,460 spots). Colors indicate the predicted tumor cell content score. The same clusters of peritumor spots in **b** are highlighted by blue dashed lines. **e** The colocalization of CNV and H3K27M mutation in spots from DMG samples (*n* = 5). The number of spots from each sample is indicated.

*ANGPT2, CD34, VEGFB*); Module 3 enriches neuron and synapse associated genes (e.g. *NEUROD1, ZIC1, CAMK2B, GABRD, SYN1*); Module 4 enriches genes involved in hypoxia and stress responses (e.g. *LDHA, HMOX1, PGK1*).

We next sought to determine the clinical relevance of these modules. We analyzed the relative module gene expression (module score) in each tumor sample, and found that high module scores are largely correlated with specific histopathological features, which were

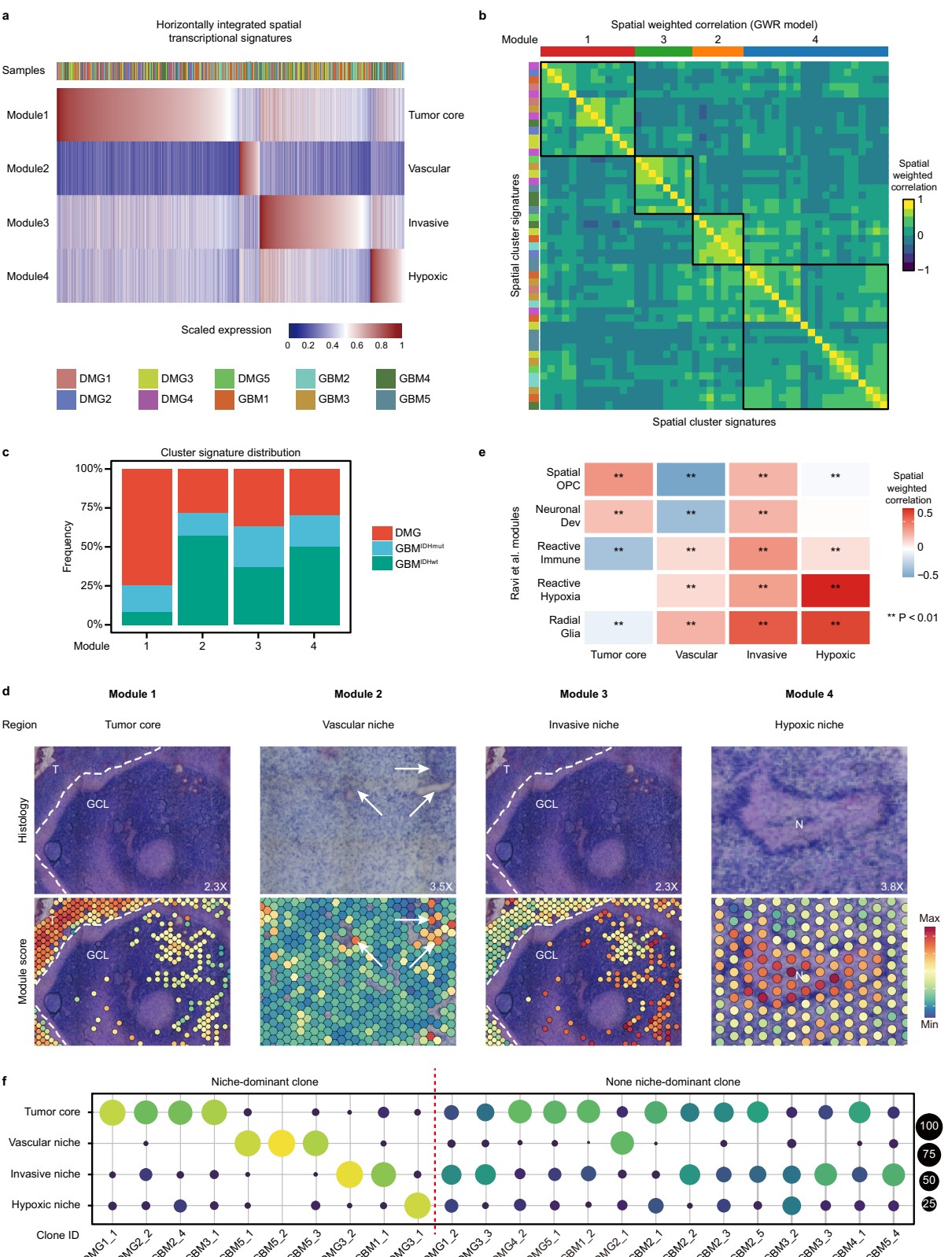

**Fig. 2 | Niche-specific spatial transcriptional programs and their relationship to subclonal architecture. a** Heatmap of horizontally integrated spatial transcriptional signatures/modules expressed in all malignant spots (*n* = 19,767). The sample identify for each spot is shown by different colors in the top bar. **b** Heatmap of GWR-based, spatially weighted correlation of the 48 spatial cluster signatures, marked by different colors on the left. Hierarchical clustering confirmed 4 clustered programs that correspond to the four modules in **a**, marked by different colors on the top. **c** The percentages of cluster signatures per tumor type contributing to each program are shown. **d** Representative examples of the correlation between the module gene expression and the histopathology of glioma niches across all samples (*n* = 10). **e** Heatmap of spatial weighted correlation between our modules and Ravi et al.[33] modules across all malignant spots (*n* = 19,767). Correlation and *P* values were determined by non-parametric Spearman's correlation test, Bonferroni-corrected for multiple comparisons. Source data are provided as a Source Data file, and individual *P* values are included. **f** Dot plot indicating subclonal distribution across different glioma niches marked by transcriptional modules from all subclones (*n* = 24). Dot size and color reflect the percentage.

further confirmed by pathologists (Fig. 2d, Supplementary Figs. 4a and 5a). Consistent with Gene Ontology enrichment analysis, Module 1 marks the tumor core with the highest tumor cell density (thereafter, tumor core). Module 2 identifies the blood vessels within and adjacent to the tumor core area (thereafter, vascular niche or vascular). Of note, GBM5 has been diagnosed as a highly vascularized tumor and the majority of the malignant spots are marked by this module. Module 3 identifies regions with relatively normal histopathology or low tumor cell density, reminiscent of tumor-infiltrated brain areas (thereafter, invasive niche or invasive). Module 4 marks the necrotic and hypoxic regions within the tumors (thereafter, hypoxic niche or hypoxic). Pseudopalisading necrosis, a hallmark feature of malignant gliomas most evident in DMG5 and GBM4, expresses high level of Module 4 geneset. Based on the relative expression of different module scores, we can assign the spots to different niches, which largely correlates with histopathology (Supplementary Figs. 4a and 5a). To test whether the spatial transcriptional modules are influenced by low-quality spots (particularly those in the hypoxic/necrotic area), we compared the gene number and mitochondrial gene percentage in spots from hypoxic niches with spots from other niches across all glioma samples, and observed statistically significant but no dramatic differences based on the median values (Supplementary Fig. 6a–c).

Ravi. et al. recently published a spatial transcriptomic dataset of 20 GBM patient samples, revealing five recurrent spatial transcriptional programs[33]. To compare our results with a larger cohort, we performed spatially weighted correlation between our pan-glioma modules and their transcriptional programs in our dataset (Fig. 2e). Our "Hypoxic niche" strongly correlates with the "Reactive Hypoxia" program. "Tumor core" correlates with "Spatial OPC" and "Neuronal development", while "Invasive niche" correlates best with "Radial glia". "Vascular niche" does not appear to have a clear counterpart, exhibiting correlation with "Radial glia", "Reactive hypoxia", and "Reactive immune". To make a direct comparison between IDH-wildtype GBMs, we further performed spatially weighted correlation between our GBM[IDHwt] modules and Ravi. et al. transcriptional programs, and found similar results (Supplementary Fig. 3b). Thus, our analysis identified similar but different spatial transcriptomic programs, possibly due to different spot filtering and data processing methods.

Malignant gliomas often exhibit spatially segregated subclonal architecture influenced by environmental stress response[33,34]. To investigate to what extent tumor subclonal architecture contributes to spatial transcriptional programs, we identified subclones in each sample based on tumor signature CNV events (total 24 subclones), and analyzed their spatial distribution across glioma niches ("Methods"). We found 10/24 niche-dominant subclones (defined as more than 75% of spots per subclone in a niche), and 14/24 non-niche-dominant subclones (Fig. 2f, Supplementary Figs. 4b and 5b). Thus, similar to the Ravi et al. study[33], subclonal architecture does not appear to play a major role in specifying the transcriptional programs.

### RNA transcript isoform diversity across glioma niches
To determine the RNA isoform diversity across glioma niches, we employed the Oxford Nanopore Technology (ONT) platform to perform long-read sequencing on the spatially barcoded cDNAs generated by 10X Visium, identifying novel isoforms as well as isoforms validated by short-read sequencing (Fig. 3a, Supplementary Figure 7a, and "Methods"). Since short-read and long-read sequencing share the same barcodes, we can align the expression of individual RNA isoforms with their spot and niche identities (Fig. 3a). Through differential expression analysis for each detected isoform, we identified isoforms enriched in different glioma niches that exhibit recurrent expression patterns across glioma samples (pan-glioma) or specific tumor types (DMG, GBM[IDHwt], and GBM[IDHmut]) (Supplementary Data 5). The host genes for these isoforms enrich for pathways correlated with their niche identities (Fig. 3b). For genes with multiple detected isoforms,

we calculated the percent spliced in (PSI) value for each isoform in every high-quality spot, and compared their PSI across glioma niches (Fig. 3a and "Methods"). We detected all types of AS events in each niche, whose percentages are not significantly different (Supplementary Figure 7b). Isoforms from the same gene but enriched in different niches are considered switched isoforms (Supplementary Data 5).

We further confirmed the AS events across glioma niches in individual samples. For example, CHI3L1 encodes a secreted glycoprotein, YKL-40, which is implicated in angiogenesis and glioma progression[35], and has been shown by Ravi et al. as a "Reactive immune" marker[33]. We detected both long (CHI3L1-201) and short (CHI3L1-205) isoforms of CHI3L1, which are differentially enriched in the hypoxic niche versus the invasive niches, highlighting the need to quantify gene expression at the isoform level (Fig. 3c–e).

To investigate the clinical relevance of niche-enriched isoforms, we compared our data with the TCGA-GBM transcription dataset[36,37]. Since the TCGA dataset is based on short-read sequencing which does not offer accurate full-length isoform information, we compared these datasets at the splicing junction (SJ) level ("Methods"). We identified 76,899 new SJs through long-read sequencing, while the remaining two thirds of SJs detected in our dataset were identified in the TCGA dataset, confirming the fidelity of our long-read sequencing (Fig. 3f). Among the shared SJs, we filter out SJs unique to specific isoforms to perform survival analysis. For example, long (Tomm6-202) and short (Tomm6-201) isoforms of Tomm6 (translocase of outer mitochondrial membrane 6) are differentially enriched in the hypoxic niche versus the invasive niches (Fig. 3g, h). The long isoform Tomm6-202 contains a unique SJ "chr6:41789337-41789530:+", whose high expression is correlated with favorable prognosis (Fig. 3i and Supplementary Data 6). To predict the regulatory mechanisms for niche-specific isoforms, we further identified splicing factors (SFs) whose expression patterns are consistent with the short isoform ("Methods", Supplementary Data 5). The AS for Tomm6 is likely regulated by YBX3, which is confirmed by eCLIP and shRNA knockdown data from the ENCODE database[38] (Fig. 3j). Consistently, low expression of YBX3, corresponding to high expression of the long isoform Tomm6-202, is associated with favorable prognosis (Fig. 3j, k). As another example, we identified the U2AF2-regulated AS of SNHG6 201/203, which has been implicated in the progression of hepatocellular carcinoma[39], and associated with patient survival (Supplementary Fig. 7c–f). Thus, our data provide a spatial profiling of diverse RNA isoforms in glioma samples, and identified survival related isoforms/SJs and potential regulatory SFs.

### Glioma ecosystem profiling reveals niche-specific enrichment of RG-like cells
Next, we sought to profile the multicellular ecosystems within different glioma niches.

We integrated Bhaduri et al. GBM scRNA-seq[8], Nowakowski et al. human cortex scRNA-seq[40], Filbin et al. DMG scRNA-seq[6], and Aldinger et al. human cerebellum snRNA-seq datasets[41] as reference datasets to perform deconvolution analysis ("Methods"). Malignant cell types from tumor datasets are named as **-like (such as AC-like), to distinguish them from cell types from normal brain datasets (such as AC). We determined the cellular composition of each spot, and calculated the average cellular abundance in spots from different niches (Fig. 4a and "Methods"). All four niches have a significant portion of astrocytes (ACs), which may be contributed by AC-like tumor cells and/or reactive astrocytes in the microenvironment, and AC-like cells are most abundant in the invasive niche (Fig. 4a). As expected, vascular and immune cells (Pericytes/endothelial cells, microglia/macrophage, and B cells) are enriched in the hypoxic and vascular niche (Fig. 4a). We found that OPCs are most abundant in the tumor core (Fig. 4a), while the neuron content is the highest in the invasive niche (Fig. 4b).

Interestingly, we found RG-like cells in both DMG and GBM spots, which are enriched in the invasive niche across glioma samples

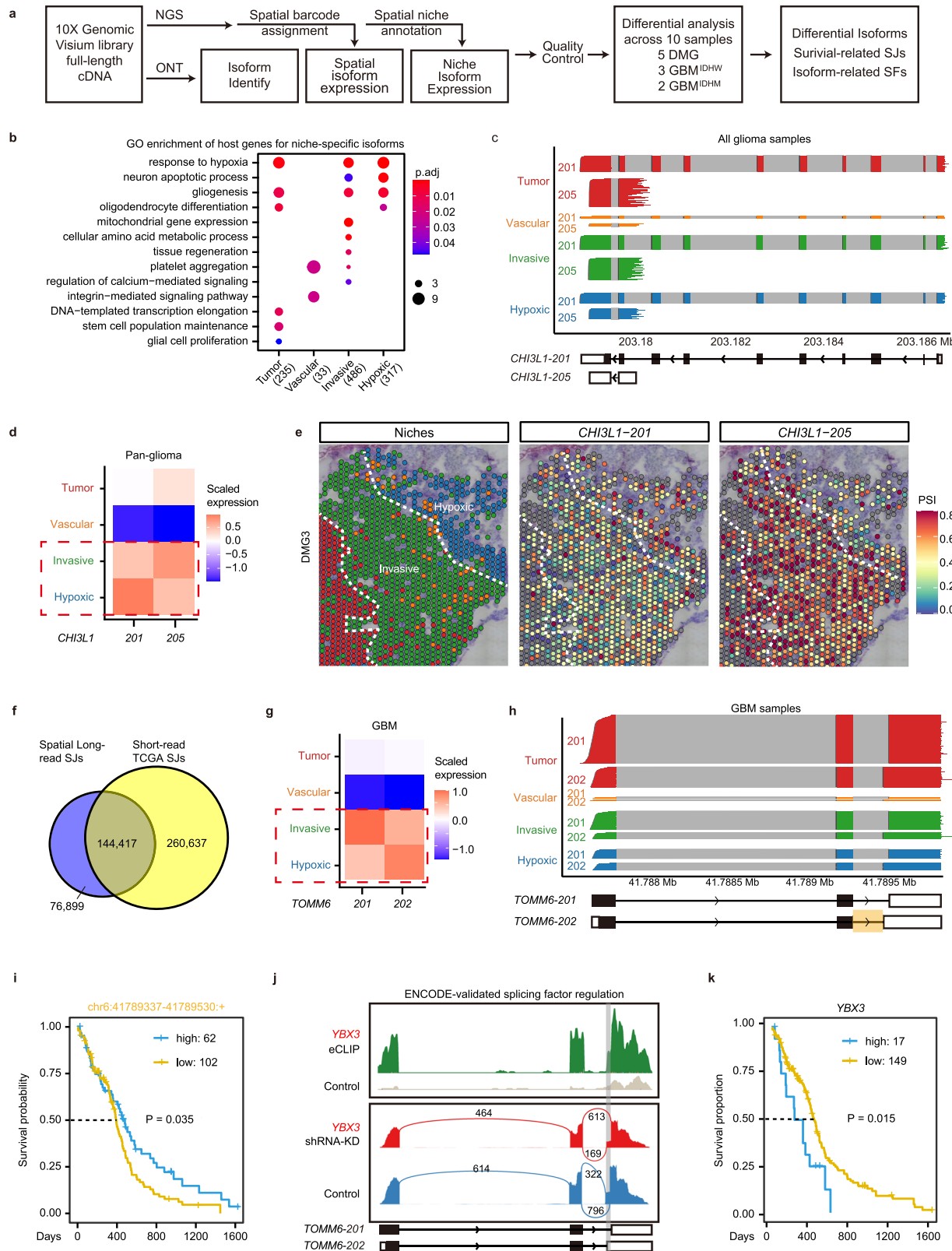

(Fig. 4a, c). In the two GBM[IDHmut] samples, hypoxic niche appears to contain even more RG-like cells than the invasive niche (Fig. 4c), although more samples are required to confirm this observation. Since it has not been reported whether DMG contains RG-like tumor cells, we rigorously tested this finding using a published DMG scRNA-seq dataset[6]. We reanalyzed the data, and found two malignant cell clusters that were annotated by the previous study as AC-like cells (AC1 and

AC2) (Fig. 4d). We found that AC2 expresses the highest level of RG signature genes among all cell clusters, suggesting its RG-like cell identity (Fig. 4e and "Methods"). It is well known that RG and astrocytes share many marker genes. To further test whether AC2 more closely resembles RG-like cells, we obtained a short list of RG-specific genes that are expressed at higher levels in RG-like cells than AC-like cells in the GBM scRNA-seq dataset ("Methods", Supplementary

**Fig. 3 | RNA isoform diversity across glioma niches revealed by long-read spatial transcriptomics. a** Workflow to identify niche-specific isoforms and splicing junctions (SJs), predict regulatory splicing factors (SFs), and analyze their prognostic value for patient survival. **b** Gene Ontology (GO) enrichment of the host genes of niche-specific isoform across samples (*n* = 10). *P* values were determined by two tailed hypergeometric test. Significantly enriched (Benjamini–Hochberg-adjusted *P* < 0.05) biological processes are shown. **c**–**e** Differential isoform enrichment of *CHI3L1* in different niches. **c** Individual transcripts of *CHI3L1* isoforms 201 and 205 in high-quality malignant spots from all glioma samples (*n* = 19,071 spots) are represented by transcript tracks, colored by their niche identities. Transcript structures based on Ensembl annotation are shown at the bottom, with colored regions representing exons and gray regions representing introns. **d** Tile plots showing scaled mean expression of *CHI3L1-201* and *CHI3L1-205* across niches to compare their niche-specific enrichment patterns. **e** As an example, spatial plots of PSI values (colors) in DMG3 quantifying the relative expression of *CHI3L1-201* and *CHI3L1-205* are shown. **f** Venn diagram showing the overlap of splice junctions detected in our long reads sequencing dataset and the TCGA-GBM dataset[37]. Differential isoform enrichment of *TOMM6-201* and *TOMM6-202* in different niches of GBM samples (*n* = 6727 spots), shown by tile plots (**g**) and individual transcripts in each niche (**h**). **i** Kaplan–Meier survival curves comparing the overall survival in the TCGA-GBM cohort[36,37], stratified by high (*n* = 62) vs. low expression (*n* = 102) of *TOMM6-202* specific splice junction. Curve comparison *P* value was determined by two tailed Log-rank (Mantel–Cox) test. **j** Predicted splicing regulation of *TOMM6* by splicing factor *YBX3*. Peaks indicate eCLIP and shRNA-KD RNA-seq read density in HepG2 cell line from ENCODE[38]. Biological replicates have similar results. Gray box shows the alternatively spliced region of *TOMM6-202*. **k** Kaplan–Meier survival curves comparing the overall survival between *YBX3* high (*n* = 17) versus low groups (*n* = 149) in the TCGA-GBM cohort[36]. Curve comparison *P* value was determined by two tailed Log-rank (Mantel–Cox) test.

Data 7). We found that the expression of these RG-specific genes is significantly higher in AC2 than AC1 cells (Fig. 4f), further supporting the presence of RG-like cells in DMG. To further confirm the relationship between RG-like cells and AC-like cells, we reanalyzed the Neftel et al. GBM dataset[7], and found that RG scores are highest in a subset of cells at the AC-like and MES-like states (Fig. 4g). Since the MES-like cell state is associated with glioma invasion but not a specific cell type[7,36,42], it is not surprising that RG-like cells exhibit MES-like state. Thus, these analyses support that RG-like cells are present in both DMG and GBM, which were classified as AC-like or MES-like cells in previous studies.

To predict niche-specific cell-to-cell communications networks, we identified 101 receptor-ligand pairs that are co-expressed in spots within each niche ("Methods" and Supplementary Data 8). Considering that there is a significant overlap of receptor/ligands expression in different cell types in brain tumors and each spot contains multiple cell types, we cannot definitively assign a receptor or ligand to a specific cell type. Interestingly, 63/101 pairs contain at least one ligand or receptor that are among RG signature genes (Supplementary Data 8).

### Regulatory programs of RG-like cells in a neuron-rich invasive niche

Since RG-like cells and neurons are enriched in the invasive niches of both DMG and GBM, we sought to determine whether there are specific regulatory programs of RG-like cells in a neuron-rich invasive niche. We focused on DMG1, a DMG sample from the cerebellum exhibiting a continuum of tumor cell infiltration from the tumor core, leading edge (LE), to a distant location in the granular-neuron cell layer (GCL_TI), with large areas of anatomically comparable normal GCL (GCL_N) that can be used as control (Fig. 5a). Consistent with the aforementioned analysis, GCL_TI as a part of the invasive niche has the highest RG and neuron scores among all malignant regions (Fig. 5b, c').

We used SPATA2 to infer the tumor invasion spatial trajectory from the tumor core towards GCL_N[43] (Fig. 5d and "Methods"). Along the trajectory, we identified genes specifically upregulated in each region based on their dynamic expression patterns. As expected, OPC markers *OLIG2*, *SOX9*, and cell cycle genes *CDK1*, *MKI67* are highest in the tumor core (early peak), while neuronal markers *ZIC1*, *GABRA1*, and mature oligodendrocyte marker *MBP* are highest in the GCL_N (late peak). GCL_TI specifically upregulates RG markers *TNC*, *HOPX*, *PTPRZ1*, and *VIM*, astrocyte markers *CLU*, *LGALS1*, as well as genes involved in migration and glioma network formation such as *CD44*, *SPARC*, and *GAP43* (One peak) (Fig. 5d and Supplementary Data 9). These genes are collectively termed GCL_TI signature.

Intersectional analyses revealed that one-fourth (*n* = 73) of the GCL_TI signature genes overlap with marker genes for RG-like cells (Fig. 5e, Supplementary Data 9), and enrich for pathways associated with interaction between tumor cells and the microenvironment cells, such as extracellular matrix organization, post-translational protein modification, synapse organization/axonogenesis, and response to

inflammation (Fig. 5f, Supplementary Data 10). Furthermore, high expression of these genes is associated with poor prognosis in the TCGA-GBM cohort (Fig. 5g). These results suggest that these genes may play a role in regulating RG-like cells in a neuron-rich microenvironment in the invasive niche.

### FAM20C mediates invasive growth of RG-like cells in a neuron-rich microenvironment

Among the top 10 upregulated genes in GCL_TI that are also expressed by RG-like cells, we found FAM20C of particular interest. FAM20C is a Golgi-localized kinase that generates the majority of the secreted phosphoproteome, and depletion of FAM20C in a breast cancer cell line impairs its migration and invasion in vitro[44]. However, its function in DMG has not been reported. The expression of FAM20C is highest in the GCL_TI of DMG1 with high neuronal content, and higher in the invasive niche than in the tumor core across glioma samples (Fig. 6a, b). In addition, its high expression is associated with poor prognosis in the TCGA-GBM cohort (Fig. 6c). These results suggest that FAM20C may play an important role in promoting the invasive growth of RG-like cells in a neuron-rich microenvironment.

To test this hypothesis, we established a DMG mouse model through orthotopic xenograft of iCas9 human embryonic stem cell (ESC)-derived neural stem cells (hNSCs)[10,45]. Through CRISPR-mediated genome editing and lentiviral infection, we established mutant hNSCs with *PDGFRA D842V*, *H3K27M* overexpression and *TP53* mutation, along with mCherry as a tracing reporter (thereafter, HPT-hNSCs) (Supplementary Fig. 8a, b and "Methods"). This oncogenic driver combination is frequently found in DMG, and was previously shown to be sufficient to drive gliomagenesis in vivo[46,47].

Since ESC-derived NSCs closely resemble RG in the developing embryonic cortex, we first investigated the function of FAM20C in HPT-hNSCs in vitro. We generated two *FAM20C* knockout HPT-hNSC lines using gRNAs targeting the translation start site of FAM20C (F1 and F2), as well as a control line using a gRNA targeting the LacZ sequence (Supplementary Fig. 8b–e). We further confirmed HPT cells are transcriptionally closest to RG-like cells in DMG and GBM scRNA-seq datasets[6,8], and expressed NSC/RG markers hNESTIN, PAX6, and SOX2 (Supplementary Fig. 8f, g). The colony-formation capacity of the F1 line is not significantly different from the LacZ control, while the F2 line shows a modest reduction (Supplementary Fig. 9a). To model the cell-to-cell communication between RG-like cells and neurons, we first performed transwell migration assay and direct coculture experiment, using LacZ, F1, and F2 HPT-hNSCs along with primary mouse cortical neurons. The hNSCs were cultured in growth factor deprived medium, whose number barely increased during the experimental interval (48 h) (Supplementary Fig. 9b). In the transwell migration assay, the number of hNSCs that migrated through the membrane towards cultured neurons is significantly reduced in both F1 and F2 groups (Supplementary Fig. 9c). During direct coculture with neurons, the number

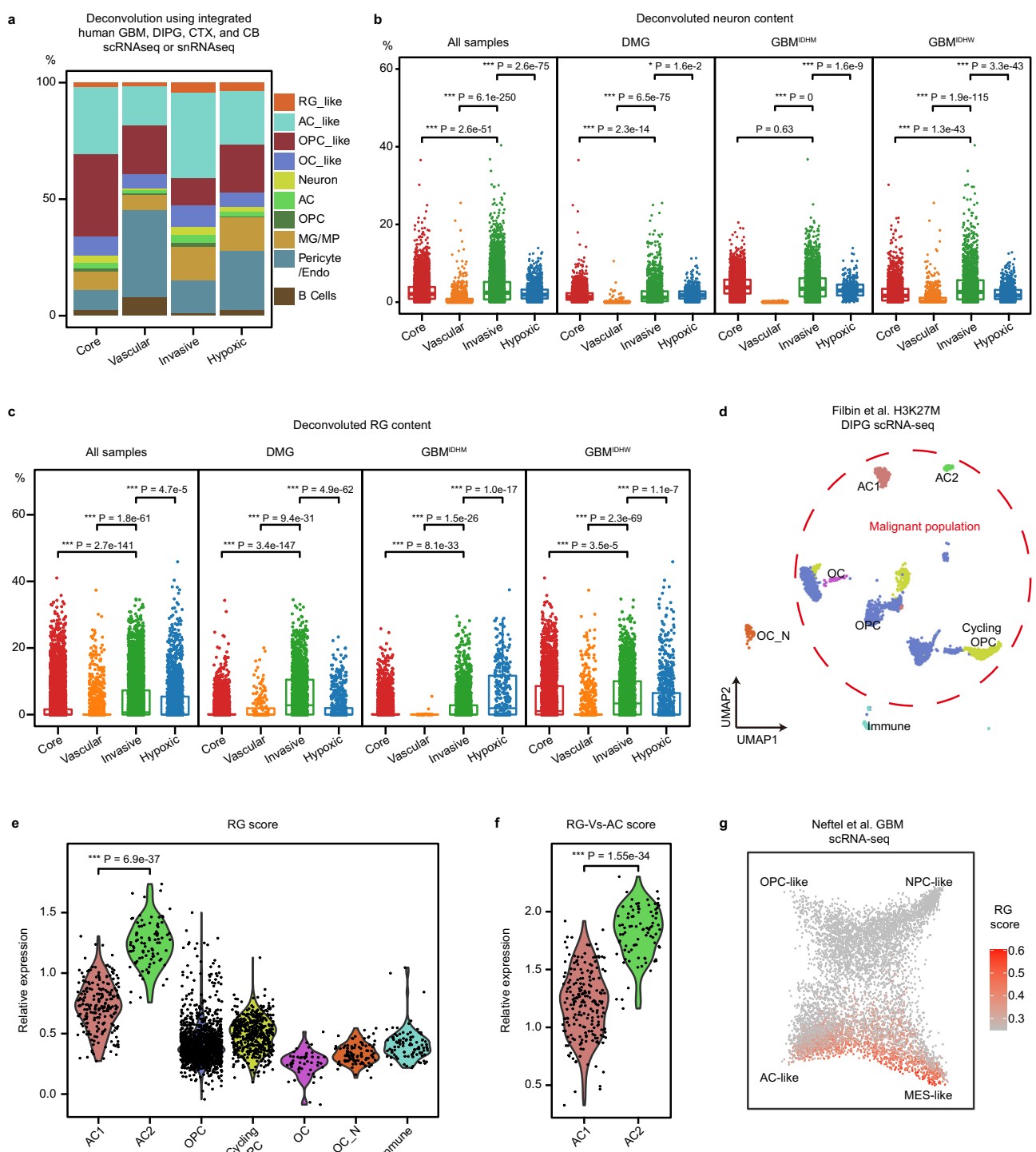

of hNSCs that grow near neuronal clusters is also reduced in F1 and F2 groups (Supplementary Fig. 9d). To test whether FAM20C regulates migration or motility of hNSCs in general, we performed a standard transwell migration assay using 10% FBS as a chemoattractant in the bottom well, but did not observe any difference in the *FAM20C* KO HPT-hNSCs (Fig. 6d). To determine whether the phenotypes we observed is caused by murine neuron-specific ligand/receptor interactions, we obtained human neurons by differentiating hNSCs in the NSC differentiation medium for 40 days to allow neuronal differentiation and maturation, as evidenced by neurite growth and expression of neuronal markers TUJ1 and MAP2 (Supplementary Fig. 9e, f and "Methods"). We re-performed transwell migration assay

and obtained consistent results (Fig. 6e). These results indicate that FAM20C promotes directed migration and growth of RG-like cells towards neurons.

To investigate the role of FAM20C in vivo, we transplanted LacZ and F1 HPT-hNSCs into the brainstem (pons) of NOD-SCID mice, a region with a large number of neurons and nuclei where DMGs commonly originate. The LacZ group developed large tumors 2–3 months after xenograft (10/10). Similar to previously reported hNSC-derived DMG models[6,46], these tumors are formed inside brain regions of pons, medulla, and midbrain, with encasement of the basilar artery, microcystic changes, and subarachnoid/subventricular spread outside the brain parenchyma, which is often seen in patients with DMG[2,46,47]

**Fig. 4 | Profiling ecosystems in glioma niches. a** The average of the deconvoluted percentage of individual cell types in spots from each glioma niche (n = 19,767). Integrated GBM[8], DIPG[6], CTX[40], and CB[41] public scRNA-seq datasets were used as a reference in the deconvolution analysis. **b** Dot plot of deconvoluted percentage of neurons in individual malignant spots from different glioma niches, quantified in all tumor samples or by tumor type (n = 10,342, 1182, 6290, and 1951 spots; n = 3917, 211, 1796, and 774 spots; n = 3713, 48, 2533, and 347 spots; n = 2712, 923, 1961, and 830 spots, respectively). Boxes indicate quartiles, horizontal bar indicates median, and whiskers indicate range, up to 1.5-fold inter-quartile range. P values, Kruskal–Wallis test for between group differences with Holm's correction for multiple comparisons. **c** Dot plot of deconvoluted percentage of RG-like cells in individual malignant spots from different glioma niches, quantified in all tumor samples or by tumor type (n = 10,342, 1182, 6290, and 1951 spots; n = 3917, 211, 1796, and 774 spots; n = 3713, 48, 2533, and 347 spots; n = 2712, 923, 1961, and 830 spots,

respectively). Boxes indicate quartiles, horizontal bar indicates median, and whiskers indicate range, up to 1.5-fold inter-quartile range. P values, Kruskal–Wallis test for between group differences with Holm's correction for multiple comparisons. **d** Re-analysis of Filbin et al. DIPG scRNA-seq dataset[6] presented by UMAP scatter plot. Cell identities for each cluster are consistent with the original study. The red-dashed circle highlights the malignant cell populations. OC_N, normal oligodendrocytes. **e** Violin plots of the relative RG score expression in individual cell types in **d**. n = 206, 87, 1437, 480, 49, 94, and 96 cells, respectively. Two tailed Wilcox test for between group differences for comparisons. **f** Violin plots of the relative RG-Vs-AC score expression in AC1 (n = 206 cells) and AC2 (n = 87 cells). Two tailed Wilcox test for between group differences for comparisons. **g** Re-analysis of Neftel et al. GBM scRNA-seq dataset[7] in scatter plot showing the expression of RG score. The cellular states for each single cell were obtained from the annotations by Neftel et al[7].

(Fig. 6f). The mCherry+ tumor cells maintain high level expression of H3K27M (Supplementary Figure 9g). FAM20C expression is most evident in the leading edge and peritumor areas with a large number of neurons, but relatively low in the tumor core and tumors outside the brain parenchyma, consistent with the observations in spatial transcriptomic analysis (Fig. 6g).

The F1 group (n = 10) became moribund at a similar rate as the LacZ group. These mice also developed subarachnoid/subventricular tumor masses outside the brain parenchyma (neuron-free), whose sizes are comparable to the LacZ group (Fig. 6f). Like the LacZ group, these mice exhibit severe hydrocephaly and enlargement of ventricles at the moribund stage, likely due to the blockade of the normal flow of cerebrospinal fluid by the tumor mass (Fig. 6f). However, inside the brain we only observed diffusely infiltrating mCherry+H3K27M+ human cells without forming large tumor mass (10/10) (Fig. 6f and Supplementary Fig. 8f). Importantly, inside the brain (neuron-rich) the percentage of Ki67+ cells among total mCherry+ cells is markedly reduced in the F1 group versus the LacZ group (Fig. 6h). As an internal control, outside the brain (neuron-free) the Ki67 percentage is comparable between these two groups (Fig. 6h), consistent with the in vitro colony-formation assay.

To rule out the possibility that the tumors outside the brain are results of continuous growth/invasion of endstage tumors inside the brain, we examined brains from three early-stage mice 21 days after HPT cell xenograft. At this stage, we can already identify clusters of mCherry+ cells in the 4th ventricle and the subarachnoid space, likely resulting from the spread of mutant hNSCs along the injection/injury path. In the pons we only observed diffusely infiltrating mCherry+ cells, which are not yet full-blown tumors and do not exhibit obvious connections with cells outside the brain (Supplementary Fig. 9h). These data indicate that from the early stage on tumors inside and outside are growing independently under different microenvironmental context. Thus, it is most appropriate to treat them as different tumor entities. Together, these in vitro and in vivo results indicate that FAM20C is required for invasive growth of RG-like cells in a neuron-rich microenvironment, but is dispensable for their neuron-free subarachnoid/subventricular growth.

## Discussion

A comprehensive characterization of glioma niches in a spatial context has the potential to improve the diagnosis and treatment of DMG and GBM. In this study, we integrated short- and long-read spatial transcriptomics as well as public single-cell transcriptomic datasets to provide a comprehensive spatial profiling of DMG and GBM, and revealed niche-specific glioma ecosystems and regulatory programs.

We identified four gene expression modules that are conserved across tumor samples, which specifically mark the hypoxic, vascular, invasive, and tumor core niches. The histopathology of these niches has been well documented in clinical practice. However, our previous understanding of these niches is largely based on bulk analyses of

regionally dissected tumor tissues, exemplified by the Ivy Glioblastoma Atlas Project[48]. Aside from the cellular tumor core, the proportion of tumor cells in other niches is generally low, making it difficult to dissect the contribution of tumor cells versus microenvironmental cells to bulk gene expression. In addition, these analyses lack a whole picture of the spatial architecture of malignant gliomas. Through spatial transcriptomics, inferCNV, mutation calling, and deconvolution using single-cell datasets, we provide a high-resolution landscape of the multicellular ecosystem within each niche.

Notably, our niche-specific transcriptional programs correlate with but differ from Ravi et al.'s spatial transcriptional programs[33]. In their study, they used a very stringent threshold (90% based on their tumor content estimation) to filter malignant spots, trying to identify tumor cell specific transcriptional programs in different microenvironment. However, our deconvolution analysis using public single-cell datasets demonstrate that there are a significant portion of immune and vascular cells within each tumor spot. The current 10X Visium platform lacks single-cell resolution and each cell contain multiple cells from different cell types. Thus, we believe it is more feasible to consider each spot as a multicellular ecosystem, and identify recurrent niche-specific signatures correlated with histopathology.

In addition to gene expression, it is increasingly recognized that RNA splicing plays an important role in the regulation of tumorigenesis[49,50]. Our study provides a spatial dataset of RNA isoforms of DMG and GBM using long-read spatial transcriptomics. The differential isoform expression across different niches may be a combined result of niche- and cell-type-specific splicing. Further analysis using long-read single-cell RNA sequencing in different regions could help us distinguish these two possibilities and identify functional RNA splicing events underlying gliomagenesis. Also, the biological significance of differentially expressed isoforms needs further validation from functional studies.

While there are many studies profiling GBM and its microenvironmental cells at the single-cell resolution[51–56], there are limited high-throughput analyses of DMG, and our understanding of this tumor type is still evolving. Our study shows that DMG shares many similarities with GBM including niche-specific ecosystems, despite its unique driver mutations and age of onset. While the DMG tumor core is largely comprised of proliferating OPCs, in the distant invasive niche we identify a high proportion of RG-like cells potentially serving as a cell-to-cell communication hub, which are transcriptionally similar to their counterparts in the GBM and developing neurogenic niche. This finding is confirmed by the re-analysis of a published DMG scRNA-seq dataset[6]. Additional single-cell analyses of regionally dissected DMG invasive niches could strengthen this point.

Since current 10X Visium-based spatial transcriptomics lacks the single-cell resolution, we believe it is essential to collect samples encompassing the tumor, peritumor, and normal brain areas with comparable anatomic structures in a single slide, along with scRNA-seq or snRNA-seq to pinpoint cell-autonomous regulators of tumor cells in

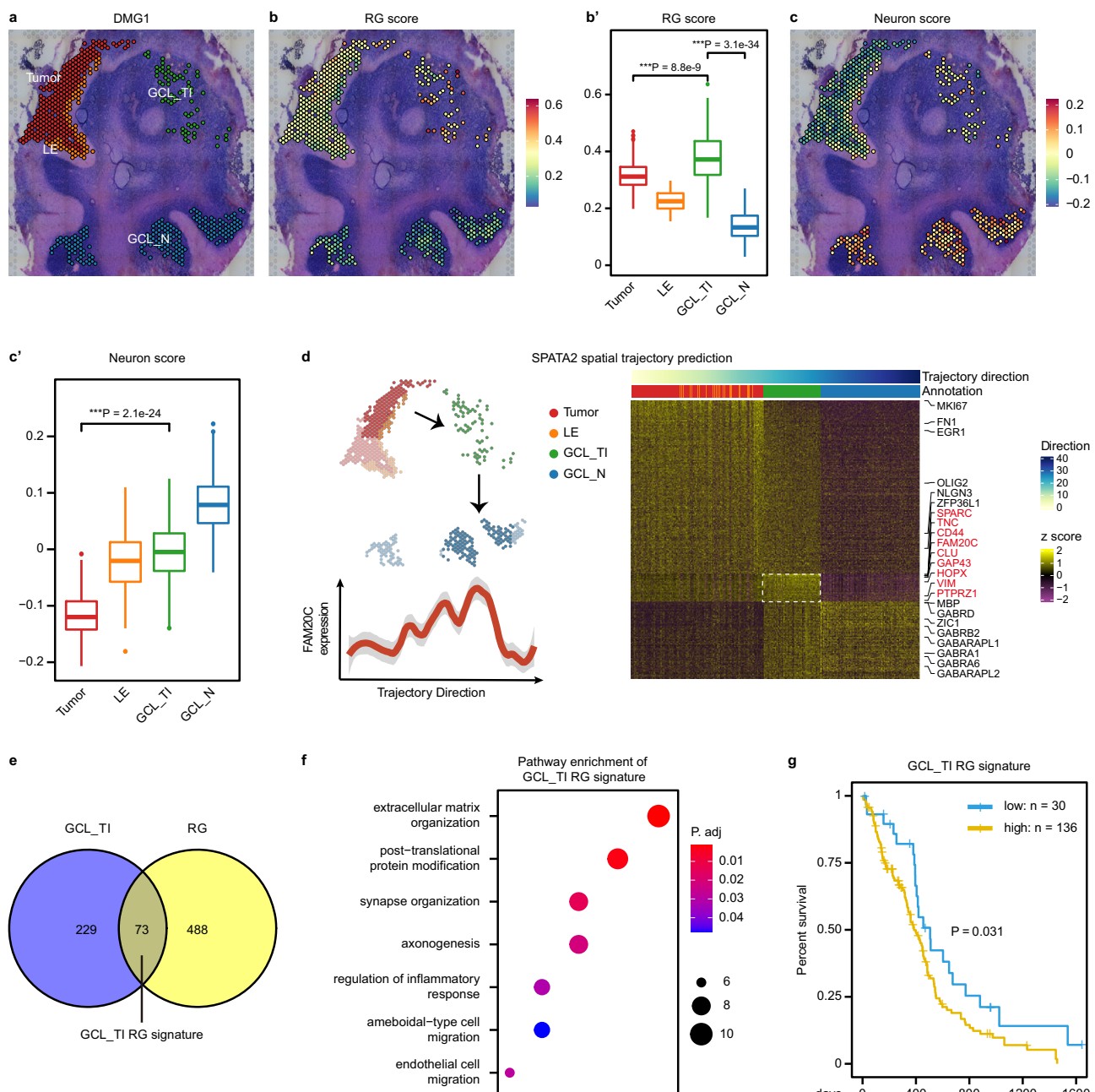

**Fig. 5 | Identification of regulatory programs of RG-like cells in a neuron-rich invasive niche. a** Selected spots from the tumor core (Tumor), leading edge (LE), and tumor-infiltrated GCL (GCL_TI), and normal spots without CNV from GCL (GCL_N) of DMG1 are shown, distinguished by color. **b** Spatial plot of the RG score in selected spots. **b′** The RG score in each region ($n = 256$, 74, 71, and 200 spots, respectively) is quantified and compared. Boxes indicate quartiles, horizontal bar indicates median, and whiskers indicate range, up to 1.5-fold inter-quartile range. P values, Kruskal−Wallis test for between group differences with Holm's correction for multiple comparisons. **c** Spatial plot of the Neuron score in selected spots. **c′** The Neuron score in each region ($n = 256$, 74, 71, and 200 spots, respectively) is quantified and compared. Boxes indicate quartiles, horizontal bar indicates median, and whiskers indicate range, up to 1.5-fold inter-quartile range. *P* values, two tailed Wilcox test for between group differences for comparisons. **d** The spatial trajectory from Tumor to GCL_N inferred by SPATA2[43]. Highlighted spots were

selected for analysis (upper left). Heatmap of genes (row) dynamically expressed in selected spots (column, marked by different colors representing regions) along the spatial trajectory is shown on the right, clustered into three modules. Genes upregulated in GCL_TI exhibiting an 'one-peak' pattern is highlighted in red. The expression of *FAM20C* along the spatial trajectory is shown as an example (bottom left). The shaded area represents the 95% confidence interval. **e** Venn chart showing the overlap between genes upregulated in GCL_TI (**d**) and the RG signature genes. The 73 overlapping genes are termed GCL_TI RG signature. **f** Gene Ontology enrichment analysis of the GCL_TI RG signature genes. *P* value was determined by two tailed hypergeometric test. Top enriched (Benjamini−Hochberg-adjusted *P* < 0.05) biological processes are shown. **g** Kaplan−Meier curves comparing the overall survival between GCL_TI RG signature high ($n = 136$) versus low ($n = 30$) groups in the TCGA-GBM cohort. Curve comparison *p* value was determined by two tailed Log-rank (Mantel−Cox) test.

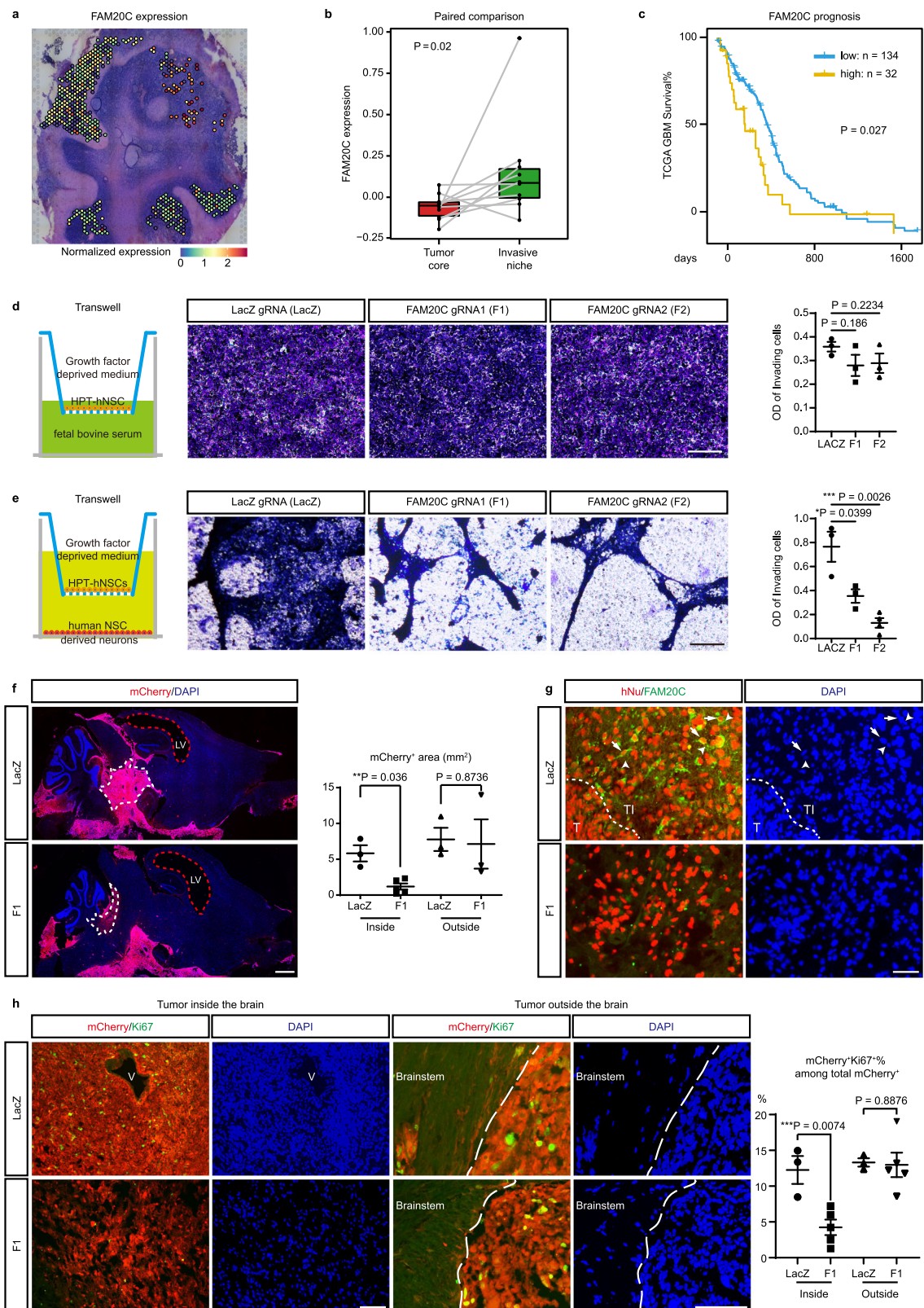

different niches. In one such sample, DMG1, we were able to simulate a whole transcriptomic trajectory of tumor spread from the tumor core to a neuron-rich invasive niche, and identify niche-specific regulatory programs in RG-like cells that are conserved across different tumors. Among the top upregulated genes, we functionally validated FAM20C as an important regulator of invasive growth of RG-like cells in a neuron-rich microenvironment, using a mutant hNSC/RG-initiated

DMG mouse model. While the exact mechanism of how FAM20C regulates the crosstalk between neurons and RG-like cells remains to be investigated, these results serve as a proof of concept that we can use spatial transcriptomics to uncover functional regulators within a specific tumor niche.

In summary, our results provide a blueprint for understanding the spatial architecture and niche-specific vulnerabilities of DMG and

**Fig. 6 | FAM20C mediates invasive growth of RG-like cells in a neuron-rich microenvironment. a** Spatial expression pattern of *FAM20C* in selected spots of DMG1. **b** The two tailed Paired Samples Wilcoxon Signed Rank Test comparing the expression of *FAM20C* between paired tumor core and invasive niches from different samples ($n = 10$). Boxes indicate quartiles, horizontal bar indicates median, and whiskers indicate range, up to 1.5-fold inter-quartile range. **c** Kaplan–Meier curves comparing the overall survival between FAM20C high ($n = 32$) versus low ($n = 134$) groups in the TCGA-GBM cohort. Curve comparison *P* value was determined by two tailed Log-rank (Mantel–Cox) test. **d** Left: Illustration of transwell assay with HPT-hNSCs in growth factor deprived medium in the top well and 10% fetal bovine serum in the bottom well. Middle: representative images of crest violet staining for migrated LacZ, F1, and F2 HPT-hNSCs. Right: Quantification of the OD values of invading cells for each group. $n = 3$ for each group. *P* values were determined by Student t-test. Error bars indicate mean ± SEM. **e** Left: Illustration of transwell assay with HPT-hNSCs in the top well and hNSC-derived neurons (day 40) in the bottom well in growth factor deprived medium. Middle: Representative images of crest violet staining for migrated LacZ ($n = 3$), F1 ($n = 3$), and F2 ($n = 4$) HPT-hNSCs. Right: Quantification of the OD values of invading cells for each group. *P* values were determined by Student t-test. Error bars indicate mean ± SEM. **f** Left: representative low-magnification images of whole-mount brain sections from LacZ and F1 group ($n = 10$ for each group). The LacZ and F1 mice shown in this panel both reached the endstage at around 80 days post HPT-hNSC xenograft. White dashed lines mark the tumor inside the brain parenchyma. Red-dashed lines mark the enlarged lateral ventricles (LV). Right: quantification of mCherry⁺ tumor area inside ($n = 3$ for LACZ group and $n = 5$ for F1 group) and outside ($n = 3$ for each group) the brain parenchyma for each group. *P* values were determined by Student t-test. Error bars indicate mean ± SEM. **g** Representative images of immunofluorescence (IF) co-labeling of FAM20C/Human Nuclear Antigen (hNu) in tumor areas inside the brains of LacZ and F1 mice ($n = 10$ for each group). The dashed line marks the border between the tumor core (T) and tumor-infiltrated brain area (TI). Arrows point to FAM20C/hNu co-labeled tumor cells in TI, arrowheads point to adjacent hNu-negative neurons with small, rounded nuclei. **h** Left: Representative images of IF co-labeling of mCherry/Ki67 in tumor areas inside and outside the brains of LacZ and F1 mice ($n = 10$ for each group). V, blood vessel. The dashed line marks the border of the brain stem. Right: the ratio of Ki67⁺mCherry⁺ cells among total mCherry⁺ cells in tumor areas inside ($n = 3$ for LacZ group, $n = 5$ for F1 group) and outside ($n = 3$ for LacZ group, $n = 5$ for F1 group) the brains of LacZ and F1 mice. *P* values were determined by Student t-test. Error bars indicate mean ± SEM. Source data for d-h are provided as a Source Data file. Scale bars, 50 µm in **d**, **e**, **g**, **h**, 1 mm in **f**.

GBM, and pave the way for future functional studies of glioma niches and ecosystems.

## Methods

### Patient samples and ethics

Our research complies with all relevant ethical regulations. Fresh tumor tissues were collected from patients with brain tumors undergoing surgical resection at West China Hospital (2020/09-2021/03). Patient sample collection and the study design was approved by the Ethics Committee on Biomedical Research of West China Hospital, Sichuan University, Chengdu, China (Approval number: 2020.837). All adult patients and the guardians of patients under 18 years old have provided written informed consent. Sample information is summarized in Supplementary Data 1.

All animal procedures were approved by the Animal Care and Use Committee of Sichuan University. Mice were housed in pressurized, individually ventilated cages (PIV/IVC) and maintained under specific-pathogen-free conditions, with free access to food and water in a 12 h light/dark cycle. The maximum allowable tumor size is 20 mm in diameter for a mouse. We confirm the maximal tumor size/burden was not exceeded, with detailed tumor diameters documented in the Source Data. When mice exhibited neurological symptoms indicative of endstage brain tumor development, or more than 20% weight loss, they were deeply anesthetized by intraperitoneal injection of Avertin (400 mg/kg) and perfused with PBS for 5 min and 4% paraformaldehyde (PFA) for 10 min. Mice were euthanized with carbon dioxide.

This study used an established human stem cell line, iCas9 hPSCs[45] and its differentiated derivative iCas9 hNSCs[10]. Its use has been approved by the Ethics Committee on Biomedical Research of West China Hospital, Sichuan University. This study does not involve establishment of new human embryonic stem cell lines, hence does not involve embryo donation or participant compensation.

### Reporting on sex and gender

Sex/gender of patients was not considered in this study, since it does not have a major impact on the spatial transcriptomes of gliomas. Sex of mice was not considered in this study, since it does not have a major impact on animal model development or xenograft growth. Female immunocompromised mice were used because they are easier to handle and pool than male ones.

### Histopathological diagnosis

Hematoxylin and Eosin (H&E) staining and immunohistochemistry (IHC) for GFAP, OLIG2, TP53, and Ki67 on tumor sections were performed by the Pathology department of West China Hospital. These slides were reviewed by two experienced neuropathologists, and tumor types were diagnosed according to 2016 World Health Organization (WHO) classification of central nervous system tumors[1]. For mutational analyses, targeted Sanger sequencing on tumor samples were performed by the Pathology department of West China Hospital using the following primers for *H3F3A*, *HIST1H3B*, *IDH1* and the promoter region of telomerase reverse transcriptase (*TERT*) gene.

H3F3A-F: 5′-GTACAAAGCAGACTGCCCGCAAAT-3′
H3F3A-R: 5′-GTGGATACATACAAGAGAGACTTTGTCCC-3′
HIST1H3B-F: 5′-CTGCTCGTAAGTCCACCGGTG-3′
HIST1H3B-R: 5′-GCGATCTCCCTCACCAACCTC-3′
IDH1-F: 5′-CGGTCTTCAGAGAAGCCATT-3′
IDH1-R: 5′-GCAAAATCACATTATTGCCAAC-3′
TERT promoter-F: 5′-GTCCTGCCCCTTCACCTT-3′
TERT promoter-R: 5′-GCACCTCGCGGTAGTGG-3′

The methylation status of O6-methylguanine-DNA methyltransferase (*MGMT*) promoter was analyzed by the Pathology department of West China Hospital through methylation-specific PCR. The primers for the unmethylated reaction were 5′-TTTGTGTTTTGATGTTTGTAGGTTTTTGT-3′ (forward) and 5′-AACTCCACACTCTTCCAAAAACAAAACA-3′ (reverse). For the methylated reaction, the primers were 5′-TTTCGACGTTCGTAGGTTTTCGC-3′ (forward) and 5′-GCACTCTTCCGAAAACGAAACG-3′ (reverse). These histopathological analyses are summarized in Supplementary Data 1.

### Sample preparation for spatial transcriptomics

**Tissue processing and sample preparation.** Surgically removed human glioma samples were snap-frozen in isopentane/liquid nitrogen, embedded in pre-chilled optimal cutting temperature compound (OCT) (Tissue-Tek O.C.T. Compound, SAKURA), and then cryosectioned into 10 µm sections at −20 °C. Sections with obvious tumor areas were collected.

**Tissue optimization.** Frozen tissue sections on Visium Tissue Optimization Slides were fixed with pre-chilled methanol. H&E staining was performed according to the Tissue Optimization Guide (CG000238 Rev A, 10X Genomics), and subjected to bright-field imaging under a Leica DM6B microscope. Sections were permeabilized with Permeabilization Enzyme at different durations. The released mRNA was captured by probes on the slides, and reverse transcribed to cDNA marked by fluorescently labeled nucleotides. Tissue was then removed from the slides with a digestive enzyme, leaving the fluorescently labeled cDNA, which was visualized under a Leica DM6B microscope. Based on the signal intensity, we determined that the optimal permeabilization duration for human brain tumor samples was 12 min.

## Spatial transcriptomic sequencing and analysis

**Visium spatial gene expression library construction and sequencing.** For spatial gene expression library preparation, frozen sections on Visium Spatial Gene Expression Slides were fixed with pre-chilled methanol and stained with H&E according to Tissue Optimization Guide (CG000238 Rev A, 10X Genomics). Sections were permeabilized with Permeabilization Enzyme at the optimal duration, and cDNAs were reverse transcribed, spatially barcoded, and amplified for 10 and 12 cycles for DMG and GBM samples, respectively.

For short-read sequencing, standard spatial transcriptomics libraries were prepared according to the Visium Spatial Gene Expression User Guide (CG0000239 Rev A, 10X Genomics). 10 µl of amplified cDNA from each sample was used for library construction through fragmentation, adapter ligation, index PCR, and purification. Each final library passed the quality control using Bio-Fragment Analyzer Qseq1 (Bioptic Inc.) and was sequenced with the Illumina NovaSeq 6000 platform at Novogene, Beijing, China, in a sequencing depth of approximately 300-400 M read-pairs per sample.

For long-read sequencing, PCR was performed with 20 ng of full-length cDNA per sample for 14 PCR cycles. PCR products were then purified and fragments greater than 800 bp were selected by 0.6X AMPure beads (Beckman Coulter, A63881). Library preparation was performed using the ONT Ligation Sequencing Kit (Oxford Nanopore Technologies Ltd., SQK-LSK109) according to the manufacturer's instructions. Approximately 200 ng of cDNA amplicons per sample were used for library preparation. Sequencing was performed on a PromethION or MinION sequencer for 72 h using R9.4.1 flow cells (Oxford Nanopore Technologies Ltd).

**Alignment and quantification.** For short-read sequencing, the sequencing data were processed using SpaceRanger software (version 1.1.0) with default parameters and mapped to the human genome (hg38 https://cf.10xgenomics.com/supp/cell-exp/refdata-gex-GRCh38-2020-A.tar.gz). Gene expression was quantified based on the unique molecular identifier (UMI). For quality control, we removed low-quality spots whose gene count <200 or the mitochondria gene ratio > 25%. To adjust for spot swapping in spatial transcriptomics data, we used SpotClean[24] (v0.99.2 with default parameters) to generate adjusted gene expression matrices for subsequent analyses.

For long-read sequencing, the raw current signal data of FAST5 files were used to obtain RNA full-length sequences using Guppy (Oxford Nanopore Technologies Ltd, v5.0.11) with super-accuracy model for base-calling. The pass sequence reads were aligned against the human genome (hg38), using minimap2[57] (v2.22 -ax splice -k14 -uf −secondary = no). We used the ScNapBar[58] (v1.0.0) software to assign standard spatial barcodes generated from short-read sequencing identification to ONT sequencing data. The parameters are set to poly A/T. Length is 10, scoring threshold is 50 and a probability model is selected. Then we used pychopper (Oxford Nanopore Technologies Ltd., v2.5.0) to filter, orient and select the full-length read for downstream analysis. For oriented strand specific reads, we used the TranscriptClean[59] (v2.0.2) to correct the mismatches, microindels, and noncanonical splice junctions for a high-quality alignment. The all corrected long reads were aligned against the human genome (hg38), using minimap2[57] (v2.22 -ax splice -k14 -uf −secondary = no). We extracted splice junctions (SJs) by comparison with GTF-annotated sites and defined novel SJs that were not annotated but supported by more than 10 reads. Then we identified all annotated and novel isoforms according to each long read and quantified the isoforms expression based on the reads from the same sample. Then, we filtered out isoforms expressed in less than 50 spots and spots expressing less than 100 isoforms by min.cells = 50 and min.feature = 100.

**CNV estimation and prediction of tumor cell content.** We applied InferCNV[11] R package (v1.7.1) following the previously described method[11], using a moving average of 100 analyzed genes to estimate CNVs in each spot and at each analyzed gene/chromosomal location. We used spots from histologically normal peritumor regions of DMG1 or GBM5_2 as independent normal references, and got similar results. CNV scores were rescaled to values ranging from 0.7 to 1.3, with scores <1 representing chromosomal loss and scores >1 representing chromosomal gain.

Since all our tumor samples exhibit broad CNVs across the genome, we utilized CNV scores to predict tumor cell content in each spot. For each sample, we first designated a tumor signature CNV event in a specific chromosomal region that is shared among spots containing tumor cells based on inferCNV and histopathology, such as Chr7 amplification in GBM, with an average CNV score >1.2 or <0.8. We then calculated the average CNV scores in this region for all spots from the same sample. The tumor content C for a given spot i was calculated as $C_i = [A_{CNVi} - 1]/[max(A_{CNV}) - 1]$ if the signature CNV is a chromosomal gain, or $C_i = [1 - A_{CNVi}]/[1 - min(A_{CNV})]$ if the signature CNV is a chromosomal loss. At least three signature CNV events were tested for each sample to ensure the robustness of the tumor content prediction. A spot is considered a malignant spot to be included in subsequent analyses if its C value is greater than 0.2.

**CNV subclone analysis.** For each sample, we clustered all spots based on their CNV profiles using average-linkage hierarchical clustering. CNV hierarchies were predicted using parsimonious order of CNV events similar to a previously published study[8]. We then split the spots into prominent clusters/subclones based on aforementioned tumor signature CNV events. The distribution of the malignant spots from a given subclone across four niches (see below *Assigning niche identities to each spot*) were quantified and compared. If more than 75% of the spots from a given subclone belong to a specific niche, we consider this subclone niche enriched.

**Calling H3K27M mutation in DMG.** We used REDItools[60] (v2) with default parameters to identify H3K27M mutation in DMG samples from BAM files generated by the SpaceRanger pipeline. The frequency of reads with H3K27M mutation in each spot is too low to be informative of the tumor cell content.

**Sample-wise spatially informed clustering.** To integrate both transcriptional space and Cartesian space for spatially informed spot clustering, we tested several recently developed spatially aware tools such as Seurat[27] (v4.0.4), SpatialPCA[29] (v1.2.0), Spruce[30] (v0.99.1), spatialDE[31] (v2), and Banksy[32] (v0.1.3). Clustered spots were overlaid on H&E images using the SpatialDimPlot function. We used DMG1 as a benchmark to compare the clustering results from these tools, and found that the clusters generated by Banksy best correlates with anatomical domains in DMG1. Thus, we used the expression matrix of each sample after preprocessing and the spatial coordinate information of the spots as input, and performed Banksy using default resolution parameters on spots from all samples to assign each spot to spatially informed clusters.

**Horizontal integration and identification of transcriptional modules across samples.** To avoid the complication from inter-patient heterogeneity, we first analyzed patient samples individually to identify spatially informed marker gene sets. For each sample, we filtered out malignant spots, performed BANKSY to group them into spatially informed clusters, and identified marker genes for each cluster using the Seurat package[27] (v4.0.4) (FindAllMarkers function, only. pos = T, p_val_adj < 0.05), while excluding marker genes that are shared by different clusters. For each cluster, we retained the top 50 marker genes based on log2FC. Clusters with fewer than 50 significant genes (log2FC > 0.25 and P.adj < 0.05) were removed. As a result, 48 spatially informed marker gene sets were identified across 10 tumor samples.

To horizontally integrate these gene sets into transcriptional modules, we tested three methods as follows and got consistent results.

(1) In the transcriptional space, we calculated the relative gene set expression score in each spot using the Seurat's (v4.0.4) AddModuleScore function with default parameters. The gene set expression matrix was then used as input for Pearson correlation analysis. The resultant correlation coefficient matrix was subjected to hierarchical clustering using corrplot package-based hclust method[61], integrating the 48 spatially informed marker gene sets into four cluster modules.

(2) In the Cartesian space, while each spot is not spatially independent, spatially informed clusters obtained by Banksy can be considered independent to each other. Thus, we integrated spots from the same cluster in each sample into pseudobulks using Seurat's (v4.0.4) AverageExpression function. For each pseudobulk, we calculated the relative expression of the aforementioned 48 marker gene sets using Seurat's (v4.0.4) AddModuleScore function with the default parameters. The gene set expression matrix was then used as input for Pearson correlation analysis. The correlation coefficient matrix was subjected to hierarchical clustering using corrplot (v0.92) package-based hclust method[61], resulting in four modules highly similar to method 1 (Jaccard-Index 0.746).

(3) In the Cartesian space, since adjacent spots are not independent, we used Geographically Weighted Regression (GWR) for correlation analysis. We first calculated all 48 marker gene set scores for individual spots in each sample. Then we calculated the spatially weighted correlation coefficient between any two gene sets using the GWmodel[62](v2.2) and gwrr[63] (v0.2-2) packages, individually for each sample. The resulting correlation array was reduced by mean to generate a single cross-sample correlation coefficient for any two gene sets. Finally, the correlation coefficient matrix was hierarchical clustered using the corrplot package-based hclust method[61], resulting in four modules similar to method 1 (Jaccard-Index 0.53). The mean values of the correlation coefficients were visualized by ComplexHeatmap[64] R package (v2.0.0).

**Spatial weighted correlation.** For spatial weighted correlation between our gene modules and published gene modules, we used GWR-based correlation as described above.

**Gene ontology (GO) enrichment analysis.** GO enrichment analysis was performed using clusterProfiler[65] (v4.6.0) R package Benjamini–Hochberg-adjusted P < 0.01 is considered statistically significant. We used dot plot function to visualize the enrichment results of GO Biological Process.

**Assigning niche identities to each spot.** We normalize the module score for each module across all spots. The niche identity of each spot was determined by its highest-expressing module score corresponding to each niche.

**Deconvolution.** We used Bhaduri et al. GBM scRNA-seq[8], Nowakowski et al. human cortex scRNA-seq[40], Filbin et al. DIPG scRNA-seq[6], and Aldinger et al. human cerebellum snRNA-seq datasets[41] as reference datasets. For each dataset, we randomly sampled 500 cells from all major cell types as input for the subsequent deconvolution analysis. Normal cells were named based on the acronyms of the cell types (such as AC), while tumor cells were named as **-like (such as AC-like). We used RCTD[66] (v1.1.0) to deconvolute the transcriptome of each spot (run. RCTD function with the entire mode parameter) into the likely constituent cell types. The frequency of the same cell types (such as AC) from different datasets were combined to obtain a single frequency for each cell type. Deconvoluted ratios of individual cell types were visualized by Seurat's (v4.0.4) SpatialFeaturePlot function.

**Identification of AS types and isoform biotype.** To identify the AS event types, a reference-guided assembly and quantification of long reads analysis was performed by StringTie2[67] (v2.1.7) with the parameter of '-L'. And then the assembled GTF was subjected to SUPPA2[68] (v2.3) with the parameter of "-f ioe -e SE SS MX RI FL" for categorizing the AS events. We classified the annotated isoforms types according to the biotype labels, and further annotated the protein-coding isoforms as principal or minor isoforms using the APPRIS database[69].

**Detection of differentially expressed isoform.** For each sample, we performed pairwise comparisons among the four niches to find niche-specific isoforms, using Wilcoxon test for differential expression testing. Differentially expressed isoforms (DEIs) were defined with the P value <0.01, absolute log2 (fold-change) >0.1 and percentage of minimal spots expression >0.1. To identify conserved isoform expression patterns across tumor samples, we only kept DEIs that were identified across patient samples with the same niche enrichment pattern. When at least two isoforms were identified as DEIs for different niches, we consider the host gene as isoform-switch genes.

**Identification of isoform-related splicing factors.** To reveal niche-specific RNA splicing networks, we selected splicing factors that might play a role in tumors[70] and performed pairwise comparisons among the four niches to find differentially expressed splicing factors (DESFs). We then overlapped the DESFs and DEIs in co-expression analysis to identify DEI-related splicing factors. Next, we downloaded the eCLIP data and shRNA knockdown data from the ENCODE database[38], and retained credible peaks (log2fold change >1 and PV >1). Reliable isoform-related SFs were found using the findOverlaps function in the GenomicRanges[71] package (v1.40.0). Finally, the shRNA knockdown data was used to verify that a splicing event is regulated by a specific SF.

**Visualization of spatial distribution of switched isoforms.** To visualize the spatial distribution of switched isoforms, we calculated the PSI of the switched isoforms in each spot according to the formula PSI = IR/(IR + ER), requiring at least 5 total reads for a gene in a spot, where IR and ER represent the reads of two differentially expressed isoforms from the same host gene. SpatialFeaturePlot function of Seurat[27] (v4.0.4) was used for PSI presentation in spatial spots.

**Visualization of long-read isoforms.** To visualize long-read transcripts, the gene region reads were extracted using Samtools[72] (v1.9). Bam files for gene were converted to BED format using Bedtools[73] (v2.29.2), and all bed files were converted to GTF format using UCSC tools bedToGenePred and genePredToGtf. Visualization was performed by R package ggbio[74] (v1.38.0). To visualize differential isoform enrichment among niches, the mean expression of an isoform in a given niche is calculated as total reads divided by the number of high-quality spots in the niche. The values from four niches were then auto-scaled and visualized in the heatmap to show niche-specific enrichment of each isoform.

**Survival analysis on TCGA-GBM cohort.** RNA-seq and clinical data of the TCGA-GBM cohort were downloaded from the TCGA Data Portal (https://tcga-data.nci.nih.gov/tcga/). For a given sample, the ssGSEA-normalized score was calculated using the GSVA[75](v1.46.0) package with default parameters based on module gene sets. The optimal cutoff point was obtained by iterating through each ssGSEA-normalized score value. And the survival curve was plotted using survminer (v0.4.9) R package. Furthermore, to explore the prognostic value of splice junctions in GBM patients, we downloaded GBM cohort in the TCGA linear junctions database and used the same approach for certain isoforms with splice junctions specific to their gene to assess the prognostic value of each splice junction.

**Analysis of public glioma scRNA-seq datasets.** Bhaduri et al. GBM scRNA-seq dataset[8] was downloaded from UCSC Cell Browser (http://gbm.cells.ucsc.edu). We obtained the list of RG marker genes from the published study[8] (p_val_adj <0.05 & avg_logFC > 0.5). We calculated marker genes expressed at higher levels in RG compared to AC using Seurat[27] (v4.0.4) (FindMarkers function: avg_log2FC > 0.5 & p_val_adj < 0.05), listed in Supplementary Data 4. We downloaded Filbin et al. DIPG scRNA-seq dataset[6] from Broad Single-Cell Portal (https://singlecell.broadinstitute.org/single_cell). We reanalyzed the data, performed cell clustering, calculated the relative expression of cell-type markers based on the original study, and assigned cell identities to each cluster. We downloaded Neftel et al. GBM scRNA-seq dataset[7] from GEO (https://www.ncbi.nlm.nih.gov/geo/query/acc.cgi?acc=GSM3828672). We calculated cell-type-specific gene set scores for both AC and RG cell types separately, and gene set scores of individual cell types were visualized by Seurat's (v4.0.4) FeaturePlot function.

**Niche-specific ligand-receptor analysis.** We extracted a pool of public ligand-receptor pairs from CellChat[76](v1.4.0) and Nichenetr[77] (v1.0.0) package. We then obtained all possible interacting pairs by pairing two genes within each module, and intersected these results with the ligand-receptor pool to predict the possible ligand-receptor pairs in each module.

**Spatial trajectory analysis of tumor invasion.** To determine the dynamics of spatial trajectories, we performed SPATA2 tool box[43] (v0.1.0) in DMG1 to manually draw trajectories simulating the tumor invasion process and selected spots included in the trajectory width 90. Dynamic gene expression changes along the trajectory were performed using the assessTrajectoryTrends function with predefined models for early peak, one peak and late peak, resulting in 3036 spatial dynamic genes. The dynamic changes of the *FAM20C* gene in the spatial trajectory were displayed by the plotTrajectoryFeatures function and the expression of spatial dynamic genes were shown in a heatmap using the ComplexHeatmap[64] package (v2.0.2).

### HPT-hNSC-derived DMG models
**Cell lines and cell culture.** The iCas9 hPSCs were gifted by Dr. Danwei Huangfu at Sloan-Kettering Institute[45], and differentiated into iCas9 hNSCs following a published protocol[10]. Briefly, iCas9 hPSCs were cultured on matrigel-coated dishes and fed daily with mTeSR (STEMCELL) for 7 days. On the next day, mTeSR was substituted by N2 medium (DMEM/F12 supplemented with 0.5× N2 supplement (Gibco), 1 µM dorsomorphin (Tocris), and 1 µM SB431542 (STEMCELL)) for 1–2 days. The hPSC colonies were lifted off, cultured in suspension on the shaker at 37 °C for 8 days to form embryoid bodies (EBs) and fed with N2 media. EBs were then mechanically dissociated, plated on a matrigel-coated dish, and fed with hNSC maintenance medium (DMEM/F12 supplemented with 1×N2 supplement, 1×B27 supplement (Gibco), 1% penicillin/streptomycin, and 20 ng/mL bFGF (Gibco)). The emerging rosettes were picked manually, dissociated completely using Accutase (Gibco), and plated on a poly-ornithine/laminin-coated plate. The resultant hNSCs were expanded and maintained in the hNSC maintenance medium.

The 293T cells were purchased from National Collection of Authenticated Cell Cultures (Shanghai, China) with catalog number GNhu177, and cultured in DMEM medium with 10% FBS and 1% penicillin/streptomycin (Gibco).

All cell lines were authenticated by the supplier. We performed additional cell line authentication by STR profiling on both iCas9 hPSCs and HEK-293T cells (TsingKe Biological Technology, Beijing, China), and confirmed their identities. Cell lines were routinely checked for mycoplasma contamination. All cell lines used in this study were tested negative for mycoplasma.

**Plasmid design.** For genome-editing, we used sgRNA sequences from published studies:[10,44,78]

sg*TP53*-5 GGGCACCCGCGTCCGCGCCA
sg*TP53*-6 GAACACTTTTCGACATAGTG
sg*FAM20C*-1 GCTGTTCGTGCGCGCAGCCC
sg*FAM20C*-2 GGCGCGCCTTGCGGGGGCGG
sg*LacZ* GCGAATACGCCCACGCGAT

These sgRNAs were synthesized by TsingKe Biological Technology (Beijing, China). For overexpression of oncogenes, coding sequence of *H3K27M* and *PDGFRA D842V* were synthesized by GENEWIZ (Suzhou, China). The sgRNAs and oncogene sequences were cloned into pLentiV2T vectors with Puromycin or Hygromycin resistance genes. Each gRNA is driven by a unique human U6 promoter. The coding sequences of H3K27M and PDGFRA D842V are driven by EF-1α-core promoter. Detailed plasmid designs are summarized in Fig. S8b.

**Lentiviral packaging and infection of hNSCs.** Lentiviruses carrying plasmids were packaged and harvested in 293T cells through calcium phosphate precipitation and concentrated by Lenti-X Concentrator (Takara). The hNSCs were cultured in coated 6-well plates, and first infected with 100 µl concentrated lentiviruses carrying V2TC-sgTP53-H3K27M-sgLacZ, V2TC-sgTP53-H3K27M-sg*F*1, or V2TC-sgTP53-H3K27M-sg*F*2. To induce the expression of Cas9, the culture medium was supplemented with 2 µg/mL doxycycline for 48 h, then screened in the culture medium containing 1 µg/mL puromycin. The genome-editing efficiency was analyzed by T7E1 or Western blot. These mutant hNSCs were subsequently infected with lentiviruses carrying V2TH-PDGFRA D842V for 8 h, then screened in the culture medium containing 100 µg/mL Hygromycin for 3 days. The expression levels of *PDGFRA* D842V in these cells were assessed by qPCR analysis.

**Western blot analysis.** The harvested cells were lysed with RIPA buffer (Sigma) containing 1 mM PMSF (Sigma). Protein samples (approximately 20 µg each) were analyzed by SDS-PAGE and electro-transferred to PVDF membrane (Millipore). The blots were then blocked in 5% skim milk in TBST, and the primary antibody was incubated overnight at 4 °C. After washing, the blots were incubated in horseradish peroxidase (HRP)-conjugated secondary antibody for 1 hour at room temperature. We used ECL or ECL Plus (GE Healthcare) to detect the signals. Primary and secondary antibodies used are as follows: β-actin (1:1000, rabbit, Cell Signaling Technology, 93473SF), FAM20C (1:500, rabbit, Abcam, ab154740), and anti-Rabbit IgG(H + L) Antibody, Peroxidase-Labeled (1:10000, ScraCare, 5220-0336).

**Confirmation of RG-like identities of HPT-hNSCs.** For IF staining of cultured hNSCs, cells were fixed with 4% PFA for 15 min at RT. After three washes with PBS, cells were treated with 0.5% Triton X-100 for 15 min at RT. After blocking with 5% milk in PBS for 1 hour at RT, cells were incubated with either anti-hNESTIN (1:2000; Abcam, ab22035), anti-Sox2 (1:1000; Abcam, ab92494), or anti-Pax6 (1:500; Abcam, ab5790), anti-mCherry (1:2000; Abcam, ab205402) overnight at 4 °C. Primary antibodies were visualized by species-specific goat secondary antibodies conjugated to Alexa Fluor dyes (Alexa 488/555/647, 1:1000, Invitrogen, A-11001/A-11008/A-21437/A-21235), and the nuclei were stained with DAPI (1 µg/mL). Stained cells were coverslipped and imaged under a Zeiss (LSM880) confocal microscope.

To determine the expression of cell-type-specific gene signatures in HPT-hNSCs, bulk RNA-seq was performed as previously described[10]. Briefly, total RNA was purified using TRIzol reagent (Invitrogen). Sequencing libraries were generated using NEB Next® UltraTM RNA Library Prep Kit for Illumina® (NEB, USA) following the manufacturer's recommendations. The library fragments were purified with AMPure XP system (Beckman Counlter, Beverly, USA). The libraries were sequenced on the Illumina HiSeq 2500 platform and 150 bp paired-end reads were generated (Novogene, Beijing, China). After QC, we aligned

the cleaned human reads to the hg38 genome for downstream analyses. We collected marker gene signatures of main glioma cell types from Bhaduri et al. GBM scRNA-seq and Filbin et al. DIPG scRNA-seq datasets using Seurat (v4.0.4) function (FindAllMarkers function, only. pos = T, p_val_adj <0.05). For a given sample, the ssGSEA-normalized score was calculated using the GSVA[75] package with default parameters based on gene signature sets. The signature score of all samples was visualized using ggplot2 R package (v3.3.6) and compared. $P < 0.05$* is considered statistically significant.

**Orthotopic xenograft.** Immunodeficient female NCG mice aged 4–5 weeks old were purchased from GemPharmatech, Ltd (Nanjing, China). For stereotactic injection, single-cell suspensions were prepared in 2 μL HBSS solution and placed on ice to maintain cells viability. $1 \times 10^6$ cells were stereotactic injected into the brainstem of NCG mice (RWD life science, Shenzhen, China). Stereotactic coordinates used were: 1 mm lateral to midline, 0.8 mm posterior to lambda suture, and 5 mm deep. The injection speed was 1 μl/min.

**Tissue preparation and Immunofluorescence (IF).** Brains were dissected, post-fixed in 4% PFA overnight at 4 °C, and then transferred to 30% sucrose overnight at 4 °C. Dehydrated brain tissues were then embedded in O.C.T. compound (Tissue-Tek) and frozen on dry ice. Cryostat sections were saggitally prepared at 10 μm thickness.

For IF staining, frozen brain sections were oven-dried at 42 °C for 30 min, rinsed and rehydrated with PBS, and treated with 0.3% Triton X-100 in PBS for 20 min at room temperature. The sections were then blocked with 2% goat serum in PBS for 1 hour at room temperature, and incubated with the primary antibody overnight at 4 °C. The primary antibodies were visualized by a species-specific goat secondary antibody conjugated with Alexa Fluor dye (Alexa 488/555/647, 1:1000, Invitrogen, A-11001/A-11008/A-21437/A-21235). Then the sections were stained with DAPI (1 μg/mL, Solarbio) for 5 min. The slides were covered and imaged under the Olympus BX51 fluorescence microscope. The primary antibodies used in this study are as follows: OLIG2 (1:1000, rabbit, Abcam, AB9610), KI67 (1:500, rabbit, BD, 550609), mCherry (1:1000, chicken, Abcam, ab205402), Human Nuclear Antigen antibody (1:200, mouse, Abcam, ab191181), FAM20C (1:200, rabbit, Abcam, ab154740). The images and statistical results presented in the figures are representative of at least three biological replicates and five different imaging fields in each group.

**Colony-formation assay.** To determine the colony-formation capacity of LacZ, F1, and F2 HPT-hNSCs, 2000 cells for each group were seeded in the coated 6-well plate and cultured until apparent colony formation. Colonies were stained by crystal violet (Beyotime biotechnology), and the total colony numbers were counted and compared. Each assay was repeated for 3 times.

**Quantitative PCR (qPCR).** Briefly, total RNA was purified from HPT-hNSCs using Trizol reagent (Thermo Fisher Scientific). Two micrograms RNA for each sample was reverse transcribed into cDNA by FastKing RT Kit with gDNase (Tiangen), prepared in iTaq™ Universal SYBR Green Supermix (BioRad), and analyzed by BioRad CFX96 Touch Real-Time PCR Detection System. Primers used for qPCR are listed below:

PDGFRA-cDNA-F: 5' TTGAAGGCAGGCACATTTACA3'
PDGFRA-cDNA-R: 5' GCGACAAGGTATAATGGCAGAAT3'
Beta-ACTIN-cDNA-F: 5' GATGAGATTGGCATGGCTTT 3'
Beta-ACTIN-cDNA-R: 5' GTCACCTTCACCGTTCCAGT 3'

**Primary mouse cortical neuron culture.** Primary mouse cortical neuron culture was performed as previously described[79]. Briefly, newborn mice at postnatal day 2 were euthanized by decapitation and the brains were dissected out. After gently stripping the meninges and

carefully separating the cortex with forceps and fine scissors, the cortical tissue was re-suspended in 2 ml of fresh dissection medium and incubated at 37 °C for 20 min. The cells were centrifuged at 4 °C at $300 \times g$ for 3 min and cell numbers were estimated by a hemocytometer, and cultured in poly-D-lysine coated plates. After incubating at 37 °C for 4 h, the old medium was removed and replaced with a fresh maintenance medium (Neurobasal supplemented with 1X B27 supplement (Gibco), 1% penicillin/streptomycin, 0.25% GlutaMAX (Gibco) and 20 ng/mL bFGF (Gibco)). When changing the maintenance medium, half of the medium were aspirated and replaced with the same volume of fresh medium every 3-4 days.

**Differentiation of human neurons from hNSCs.** We generated human neurons through direct differentiation of hNSCs using an adapted protocol[80]. Briefly, hNSCs were dissociated by Accutase and seeded into poly-D-lysine and laminin-coated 6-well plate ($2 \times 10^5$ cells /well) or 24-well plate ($1 \times 10^5$ cells/well), and cultured in hNSC culture medium (day 1). On day 2, bFGF was withdrawn and 5 μM Rock inhibitor (Y-27632, STEMCELL Technologies) was added to the medium for 24 h. After that hNSC culture medium minus bFGF was changed every other day until day 40. We performed immunocytochemistry to confirm the neuronal identity of differentiated cells. Cells were fixed with 4% PFA at RT for 15 minutes, washed with DPBS for three times, and blocked with 5% goat serum in DPBS for 1 h at RT. Cells were then incubated with anti-MAP2 (1:5000, Chicken, Novus, NB300-213) and anti-TUJ1 (1:5000, Rabbit, Sigma, 801201) overnight at 4 °C. Primary antibodies were visualized by anti-chicken or anti-rabbit goat secondary antibodies conjugated to Alexa 555 (1:1000, Invitrogen, A-21437) or Alexa 488 (1:1000, Invitrogen, A-11008), respectively, and the nuclei were stained with DAPI (1 μg/mL).

**Cell Counting Kit 8 (CCK-8) assay for cell viability/proliferation.** 1 $\times 10^4$ cells/well (200 μl cell suspension) HPT-hNSCs were placed into a 96-well plate, and cultured in normal hNSC culture medium or growth factor deprived hNSC culture medium. At 0, 24, and 48 h, 10 μl CCK-8 solution (Bimake, B34302) was added directly to each well and incubated for 3 h. Then the absorbance at 450 nm for each well was measured with a microplate reader.

**Transwell migration assay.** Cell migration was assessed by the Transwell® Permeable Supports assay (Polycarbonate Membrane, Corning) in a 24-well transwell system. (1) The standard transwell migration assay was performed by adding 600 μl DMEM medium supplemented with 10% FBS in the bottom chamber. (2) For migration towards mouse primary neurons, the bottom chamber was pre-coated with poly-D-lysine and seeded with $2 \times 10^5$ freshly isolated primary mouse cortical neurons. The neurons were cultured in maintenance medium for 10 days until they formed apparent synapses. (3) For migration towards human neurons, the bottom chamber was pre-coated with poly-D-lysine and laminin, and seeded with $1 \times 10^5$ wildtype hNSC. The hNSCs were allowed to differentiate following the aforementioned protocol for 40 days until they formed apparent synapses.

For all transwell migration assays, the top insert was seeded with 2 $\times 10^5$ HPT-hNSC with or without *FAM20C* KO on top of the permeable membrane in growth factor deprived culture medium (hNSC culture medium without growth factors). After 48 h, the HPT cells that migrated through to the other side of the membrane were fixed with 4% PFA and stained with crystal violet. The migrated cells were imaged under the microscope. The intensity of crystal violet staining was measured by ImageJ, and at least 10 imaging fields for each group were quantified and compared. To measure the OD of invading cells, the dye was collected in 800 μl 10% acetic acid (Keshi) following a published protocol[81]. For migration towards FBS, 100 μl solution of dissolved dye was further diluted 1:5 in 400 μl 10% acetic acid. For migration towards neurons, no further dilution was required. For each sample, 200 μl

solution was subjected to absorbance measurement at 552 nm, and at least three replicates for each group were quantified and compared.

**Direct coculture of mouse neurons and HPT-hNSCs.** $7.5 \times 10^5$ Primary mouse cortical neuron were cultured in coated six-well plates for 10 days until they form apparent synapses. LacZ, *F1*, and *F*2 HPT-hNSCs were dissociated into single cells using Accutase (Gibco) and seeded with the neuron for 3 days. For IF staining of co-cultured cells, cells were fixed with 4% PFA for 15 min at RT. After washed with PBS three times and blocked with 5% milk in PBS for 1 h at RT, cells were incubated with anti-mCherry (1:1000, Chicken, Abcam, ab205402) overnight at 4 °C. Primary antibodies were visualized by anti-chicken goat secondary antibodies conjugated to Alexa 555 (1:1000, Invitrogen, A-21437), and the nuclei were stained with DAPI (1 μg/mL). HPT-hNSCs clustering around densely populated neurons were imaged under the microscope and counted by Image J. The images and statistical results presented in the figures are representative of at least five different imaging fields and three biological replicates for each group.

### Statistical analysis
Statistical analyses were performed by R version 4.0.3 and Graphpad Prism (v9.0). Statistic tests and data presentation are indicated in the figure legends, and P or P adj. <0.05 is considered statistically significant.

### Reporting summary
Further information on research design is available in the Nature Portfolio Reporting Summary linked to this article.

## Data availability
The raw data for short-read sequencing in this study were deposited in Genome Sequence Archive with accession ID HRA001865. The raw data for long-read sequencing in this study were deposited in Genome Sequence Archive with accession ID HRA001960. The raw data for bulk RNA-seq of HPT cells were deposited in Genome Sequence Archive with accession ID HRA003511 (https://ngdc.cncb.ac.cn/gsa-human/browse/HRA003511). Since these data are related to human genetic resources, raw data can be obtained within 3-6 weeks by requesting and following the guidelines for Genome Sequence Archive for non-commercial use. There are no time restrictions once access has been granted. The guidance for making a data access request of GSA for humans can be downloaded at the National Genomics Data Center website (https://ngdc.cncb.ac.cn/gsa-human/document/GSA-Human_Request_Guide_for_Users_us.pdf). The processed data in this study have been deposited in the GEO database under the accession GSE194329 and Figshare (https://doi.org/10.6084/m9.figshare.20653908). The corresponding author will respond to requests for the data within one week. The TCGA GBM publicly available data[36] used in this study are available in the TCGA Data Portal (https://tcga-data.nci.nih.gov/tcga/). The Bhaduri et al. GBM scRNA-seq[8] publicly available data used in this study are available in UCSC Cell Browser (http://gbm.cells.ucsc.edu). The Filbin et al. DIPG scRNA-seq[6] publicly available data used in this study are available in Broad Single-Cell Portal (https://singlecell.broadinstitute.org/single_cell). The eCLIP and HepG2 shRNA knockdown publicly available data used in this study are available in the ENCODE Data portal[38] (https://www.encodeproject.org). The Nowakowski et al. human cortex scRNA-seq[40] publicly available data used in this study are available in UCSC Cell Browser (http://cells.ucsc.edu/?ds=cortex-dev). The Aldinger et al. human cerebellum snRNA-seq[41] publicly available data used in this study are available in Human Cell Atlas (https://www.covid19cellatlas.org/aldinger20) and the UCSC Cell Browser (https://cbl-dev.cells.ucsc.edu). The remaining data are available within the Article, Supplementary Information or Source Data file. Source data are provided with this paper.

## Code availability
The scripts used to perform the main analyses are available in Figshare (https://doi.org/10.6084/m9.figshare.21878748).

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

## Acknowledgements

We thank Bin Chen for technical support. Y.W. is supported by the National Key Research and Development Program of China, Stem Cell and Translational Research (2022YFA1105200 and 2017YFA0106500), the National Natural Science Foundation of China (82273117 and 31871376), the Distinguished Young Scientists Program of Sichuan Province (2019JDJQ0029), and the 1·3·5 project for disciplines of excellence, West China Hospital, Sichuan University (ZYYC20019). YK.Z. is supported by the Science and Technology Supportive Project of Sichuan Province (2022YFS0049). Y.J. is supported by the Science and Technology Supportive Project of Sichuan Province (2021YJ0185). Y.Z. is supported by National Natural Science Foundation of China (82173179), and the National Clinical Research Center for Geriatrics, West China Hospital, Sichuan University (Z2021JC006). L.C. is supported by the National Key Research and Development Program of China, Stem Cell and Translational Research (2017YFA0106800), and the National Natural Science Fund for Excellent Young Scholars (81722004). Y.R. is supported by the Science and Technology Supportive Project of Sichuan Province (2022YFS0143), the Technology Innovation and Development Project of Chengdu Science and Technology Bureau (2021-YF05-01038-SN), and the Postdoctoral Research Fund, West China Hospital, Sichuan University (20HXBH033). A.X. is supported by the the Science and Technology Supportive Project of Sichuan Province (2022YFS0320).

## Author contributions

Y.W. conceived the study and wrote the manuscript. Y.W., YK.Z., Y.J., L.C., and Y.Z. designed and supervised the experiments, and analyzed the data. Y.R. and P.X., assisted by T.C. and B.H., performed sample preparation and spatial transcriptomic sequencing library construction, and helped with manuscript preparation. Z.H. and J.S., assisted by L.C., P.H., and R.Z. performed most of the computational analyses, analyzed the data, and helped with manuscript preparation. L.Z., assisted by C.X., X.D., and M.L. performed most of the FAM20C experiments and analyzed the data. YK.Z., Y.J., Q.M., C.Y., J.X., Y.L., assisted by Z.L., T.Z., Q.G., Y.Y., X.Y., A.X., performed neurosurgeries and provided the samples. Z.S. and Y.O. performed histopathological examinations.

## Competing interests

The authors declare no competing interests.
