## [Peer Review File · Nature Communications]

Spatial transcriptomics reveals niche-specific enrichment and vulnerabilities of radial glial stem-like cells in malignant gliomasReviewers' Comments:

Reviewer #1:

Remarks to the Author:

This study describes the spatial transcriptomic landscapes of DMG and GBM from 10 patient tumors and one non-tumor region. They performed spatial RNA sequencing (not a single cell level analysis) and leveraged existing single-cell RNA sequencing datasets to infer cellular composition from various "niches" they defined. While the study is interesting, there are numerous interpretation and experimental design issues that significantly reduce the potential impact of this study.

Major concerns

- 1) Many conclusions are premature, overinterpreted, or selective. For example, it is stated that they "functionally confirmed that FAM20C specifically mediates gliomagenesis of RG-like cells" but this is not substantiated by the data. If anything, the authors state (but do not show) that mice transplanted with control KO and FAM20C KO cells reach moribund states at a similar rate, which is contrary to the above statement. Furthermore, one of the FAM20C KO cell lines had equivalent colony formation ability as the control KO cells; hence, the conclusion that "FAM20C promotes migration and growth of RG-like cells toward neurons (lines 282-283) is not supported by their data. Also, it is stated that "TPM2-201 is preferentially enriched in the invasive niche- line 154"; however, it is equivalently enriched in the hypoxic niche as well, but this was omitted. Similar omission for the invasive niche in line 183. In addition, line 246-247 ("these genes may play a cell-autonomous role") is not substantiated by the data presented.
- 2) Figure 3E heatmap does not seem to match the percentages shown on the right panel.
- 3) Analysis of 5 samples of each tumor type (DMG and GBM) is not sufficient to call this an "atlas" study.
- 4) The "normal region" seems to contain cancer cells as well as normal cells- as shown in Supp Fig S2. This should be acknowledged in the text.
- 5) Many observations, especially those claiming quantitative differences, should be quantified and statistical significance calculated and presented in the figure. For example, Fig 1C.
- 6) It is unclear why mouse neural cells were used in the migration assay. It is well known that there are cross-species differences in receptor/ligand interactions. They should use differentiated human NSCs instead. Also, these experiments do not address the molecular mechanisms of FAM20C function in RG-like cancer cells or the invasion niche. Does it regulate migration or motility in general or directed migration towards neurons?
- 7) The functional significance of many observations is not clear. For example, while detecting alternative splicing patterns of various genes is interesting, it is not clear how this information impacts our understanding of gliomagenesis. This should be more explicitly stated and tested in the proper context.
- 8) Considering that there is a significant overlap in expression patterns of receptor/ligands in different cell types in brain tumors (for example, cancer cells can express some traditionally considered immune cell markers), and since each spot in the spatial-seq data contains multiple cells of unknown numbers and lineages, cell-to-cell communication network analysis shown in Fig 4g is problematic.
- 9) Key references are missing in many places and previous work demonstrating the same conclusion is missing. For example, lines 134-137 state that hypoxic and vascular niches are negatively correlated with prognosis, consistent with the notion that cells within these niches may evade standard therapies... There are many publications in the last decade that demonstrated this point and should be referenced.
- 10) They should show tumor volume "outside" of the brain in control and FAM20C KO gliomas. It looks like the leptomeningeal spread is increased in FAM20C-KO tumors.

Minor concerns:

- 1) Statistical testing should be more detailed, and the numbers of replicates should be clearly indicated.

Reviewer #2:

Remarks to the Author:

In the presented manuscript, the authors use 10X Visium Spatial Transcriptomics, including Illumine short and Nanopore long read sequencing, to map the spatial distribution of recurrent transcription programs. The term "multiomics" in the title is misused because the authors only examined the transcriptome; splice variants, although highly valued, do not justify the term "multiomics." The study included 11 data sets (from 10 patients) with 5 DMG, 2 IDH-mutated, and 3 IDH-WT samples. Unfortunately, the cohort of different entities is very small and it is difficult to draw conclusions from the small sample size. My greatest concern is that the authors mixed the entities and did not analyze each entity separately. IDH-mutated and WT tumors are different cancer types as is DMG. The spatial transcriptomics technology is new and the data sets are highly appreciated. Since the authors obviously followed the preprint published last year (Ravi et al.) (similar structure of figures and analytical approach), they should discuss and compare their results with the five spatially separated programs presented previously. Including and comparing their results with a large cohort may help to support their findings. The manuscript is clearly written, but sometimes lacks detail; in particular, I was disappointed by the superficial description of the methods. The authors need to work on the methods section to ensure reproducibility.

Another high-level problem is that the authors did not use the latest computational tools developed for spatial transcriptomics while instead using tools tailored for transcript space analysis but ignoring the level of cartesian space. One example is the use of correlation: here the authors used a correlation analysis (Pearson? Not mentioned!) that considers each spot as independent, which is not correct in spatial analysis. Since adjacent spots are not independent, the authors must use correct statistical models such as GWR. In other words, the correlation must be corrected for spatial dependencies.

The authors used pseudotime estimation to study expression differences localized in cartesian space. First, pseudotime estimation is flawed (mathematically and biologically) in non-single-cell, spatially resolved transcriptomics unless the authors can correct the model based on the cellular composition of individual spots. Why don't the authors use spatial trajectory analysis to estimate gene expression gradients in space (SPATA2 toolbox ?).

Further (minor) concerns:

The authors do not mention that 10X Visium does not have single cell resolution which complicates the analysis and interpretation of the data. First, they should show how many cells are located within each spot and the diversity across samples. Using all spots for cluster analysis will lead to high confounder based on spots which are only contain a low frequency of cancer cells. CNAs are a nice marker for GBM to identify the tumor content within each spot, properly the K27 mutation (or IDH mutation) can be used. At least the authors need to discuss and address this problem.

"Unsupervised Non-negative matrix factorization (NMF) was performed on all malignant spots using the NMF R package23 (version 0.23.0)." This approach is highly biased for several reasons: 1. As the authors show in Figure 1, the data are clustered by patient, which is not surprising due to the lack of horizontal integration in the analysis!!! However, even after horizontal integration, spatially resolved transcriptomics and single-cell data (not as strong as spatial data) of cancers show high heterogeneity, called inter-patient heterogeneity, which has been recently discussed in several papers (Nefitel et al., 2019). Therefore, the community strongly recommends analyzing patients individually and then comparing clusters to identify metamodules. For spatial data, authors should also compare the spatially weighted correlation of metamodules for clustering (not just correlation as shown here).

In Figure 2 C, the spots are differently visualized (compare hypoxic vs. the rest (spot and spot

distance)?

The authors state that the hypoxic signature is enriched in necrosis. Shouldn't necrosis be excluded from cluster analysis? Necrosis can be well visualized by the total number of UMIs. What was the number of UMIs in the areas defined as necrosis? Do the hypoxic niches show a lower number of UMIs compared with the other groups? Wouldn't this be a confounding factor based on the fact that higher stress sample preparation is automatically defined as hypoxia?

Spatial data have hidden dependencies (as mentioned above). Another level of potential confounder is the spatial distribution of genetic heterogeneity or subclonal architecture (which is spatially segregated). Authors should investigate whether their transcriptional programs are not derived from subclonal architecture. Since subclones also have a specific spatial distribution, the question arises whether transcriptional programs follow the subclonal architecture or occur independently. (See Neftel et al. subgroups vs subclones...). Since the Verhaak group has shown that metabolic (hypoxia) stress leads to an increase in CNAs and epigenetic dysregulation (Johnson KC et al, 2021)

"Then the original Louvain algorithm (FindClusters) with clustering resolution 0.8 or 3.0 was performed to cluster spots and we manually combined clusters based on histology"... A good example of the lack of detail in the method. 1. Why 0.8 or 3.0 sounds very high to me. Normally an iterative approach should be taken, testing all resolutions from 0.1 to, say, 3. Then the stability of the cluster should be tested for all resolutions to find the most stable resolution. For spatial data, clustering methods that incorporate Cartesian space, such as BayesSpace, should be used in the first instance.

The additional data on splice variants is of great interest, but needs to be better explored. How different is the situation in the subgroups. Can the authors find spatial gradients of splice variants within the subgroups?

Deconvolution with 3 different data sets is very difficult to interpret. The authors should start integrating the datasets (horizontal integration / or azimuth mapping), including a cortex dataset (for the GBM). No statistics will be provided!

The discussion between AC and RG is not clear to me. The authors also need to include the Neftel classification (as a standard single cell classification). I fully agree with the authors that the RG population is a distinct cell population responsible for migration and invasion. They should show that these cells are described as part of the AC-like subset in DMG and GBM. The statement that this population has not been described is not true, it has just been named differently.

The analysis performed in Figure 5 is interesting, but the method used is partially flawed (as mentioned above). The authors should include spatial trajectories and spatially weighted correlation analyses.

The authors state that their scores based on TCGA analysis have clinical implications. Do the authors not contradict themselves by showing that their signatures are recurrent in all patients and then showing that the signatures are associated with better or worse survival? Is the percentage of signatures relevant? Does it matter how much of which subgroup was resected? There is so much room for error here that I don't think the comparison is useful.

last but not least, I would like to note that the authors need provide a QC overview with the most important parameters such as Mean UMI Max UMI and Min UMI nr. genes per spot (man min), sequencing saturation percentage of reads in sample and per spot (min max) and so on ...

Finally, I would like to motivate the authors, the data set warrants publication in a reputable journal. I am aware that it is not easy to present the data and consider all levels of complexity, but the authors

had better try. I would like to motivate the authors to understand my concerns, because honesty helps to optimize the mouse script!!!! I am looking forward to the revised version

Response to Reviewers

Response to Reviewers (NCOMMS-22-06690A)

We'd like to thank both reviewers for their positive comments on our manuscript and constructive suggestions, and we apologize for the extended revision time due to the COVID 19 pandemic. We have substantially revised our manuscript and figures, included new data, and added two more supplementary figures and two more supplementary tables, and uploaded both a clean and a track-change version of the revised manuscript.

Below, we provide a point-by-point rebuttal to the issues the reviewer raised. Page number references are based on the clean version of our revised manuscript.

Reviewer #1 writes: *“This study describes the spatial transcriptomic landscapes of DMG and GBM from 10 patient tumors and one non-tumor region. They performed spatial RNA sequencing (not a single cell level analysis) and leveraged existing single-cell RNA sequencing datasets to infer cellular composition from various “niches” they defined. While the study is interesting, there are numerous interpretation and experimental design issues that significantly reduce the potential impact of this study.”*

Response: We thank this reviewer for his interest in our study and valuable suggestions.

Major concerns:

1) *“Many conclusions are premature, overinterpreted, or selective. For example, it is stated that they “functionally confirmed that FAM20C specifically mediates gliomagenesis of RG-like cells” but this is not substantiated by the data. If anything, the authors state (but do not show) that mice transplanted with control KO and FAM20C KO cells reach moribund states at a similar rate, which is contrary to the above statement. Furthermore, one of the FAM20C KO cell lines had equivalent colony formation ability as the control KO cells; hence, the conclusion that “FAM20C promotes migration and growth of RG-like cells toward neurons (lines 282-283) is not*

Response to Reviewers

supported by their data. Also, it is stated that “TPM2-201 is preferentially enriched in the invasive niche- line 154”; however, it is equivalently enriched in the hypoxic niche as well, but this was omitted. Similar omission for the invasive niche in line 183. In addition, line 246-247 (“these genes may play a cell-autonomous role”) is not substantiated by the data presented.”

Response: We thank the reviewer for pointing out potential problems in our writing and data interpretation.

For the FAM20C in vivo experiments, the FAM20C F1 KO line, while as the reviewer pointed out did not exhibit colony formation deficit in vitro, can be used to specifically test the impact of FAM20C KO on the invasive growth of RG-like cells in vivo. However, when we analyzed the data, we encountered an obvious complication: similar to previously reported human stem cell-derived DMG/DIPG models, the CTR mice developed tumors both inside (neuron-rich) and outside the brain (neuron-free), and the moribund state may be caused by either tumor type. Indeed, we observed severe hydrocephaly and enlargement of ventricles in moribund mice from both CTR and FAM20C KO groups, likely caused by blockade of the normal flow of cerebrospinal fluid. While FAM20C KO group also developed large tumors outside the brain comparable to the CTR group (see also Response to Reviewer 1, Major Concern 10), they barely developed tumors inside the brain. Thus, FAM20C appears essential for RG-like cells in a neuron-rich microenvironment, but is dispensable in a neuron-free subarachnoid/subventricular microenvironment, which agrees with our in vitro data and bioinformatic analysis. We added this discussion in our revised manuscript text (Page 13). To avoid confusion, we have toned down and revised our abstract, section title, and manuscript text from “gliomagenesis of RG-like cells in a neuron-rich microenvironment” to “invasive growth of RG-like cells in a neuron-rich microenvironment”.

Response to Reviewers

The conclusion “FAM20C promotes migration and growth of RG-like cells toward neurons” is a summary of our in vitro experiments, which we believe is supported by the data. To address the possible cross-species differences in receptor/ligand interactions, we performed additional in vitro transwell migration experiments using hNSC-derived neurons, and got consistent results (see also Response to Reviewer 1, Major Concern 6).

Finally, we have systematically checked and revised our text to offer a more complete and balanced description of our results. Since *TPM-201* is no longer a differentially expressed isoform based on our new niche-assignment, we deleted its description in our revised Fig. 3 and text. For the sentence in *line 246-247*, we have toned down and removed “cell autonomous” (Page 11).

2) “*Figure 3E heatmap does not seem to match the percentages shown on the right panel.*”

Response: We apologize for not detailing how these percentages were calculated. A gene may have more than two alternatively spliced transcripts. The heatmap on the left was calculated as the percentage of a specific transcript among total transcripts from the same gene, i.e. $SERPINA3\ 201/(201+204+\text{other transcripts})$. The percentage on the right was calculated as the relative proportion between 201 and 204, i.e. $201/(201+204)$. While the patterns are similar, our original presentation did cause unnecessary confusion. In our revised Fig. 3 and Fig. S6, we used the track images to show the total reads and transcript structures of each isoform in different niches, while using the heatmap to compare the differential expression of isoforms in different niches.

3) “*Analysis of 5 samples of each tumor type (DMG and GBM) is not sufficient to call this an “atlas” study.*”

Response to Reviewers

Response: We apologize for the misuse of “atlas” and have systematically changed it to “profiling” or “dataset”.

4) *“The “normal region” seems to contain cancer cells as well as normal cells- as shown in Supp Fig S2. This should be acknowledged in the text.”*

Response: We agree with the reviewer. Indeed, our new tumor content estimation also identify spots with tumor cells in this sample (revised Fig. 1d). Of note, we chose spots in the histologically normal cortex region instead of the entire GBM5_2 sample as a reference for inferCNV analysis, minimizing the impact of potential cancer cells. We also repeated our InferCNV analysis using histologically normal region from DMG1, and got consistent results. We have acknowledge this in the text as follows: “Of note, the peritumor sample GBM5_2 also contained an area of spots with CNVs, which were excluded from the normal reference” (Page 4).

5) *“Many observations, especially those claiming quantitative differences, should be quantified and statistical significance calculated and presented in the figure. For example, Fig 1C.”*

Response: We have systematically revised our figures to present quantification and statistical significance. For Fig. 1C, we quantified the co-localization of H3K27M and CNV in each DMG sample in our revised Fig. 1E.

6) *“It is unclear why mouse neural cells were used in the migration assay. It is well known that there are cross-species differences in receptor/ligand interactions. They should use differentiated human NSCs instead. Also, these experiments do not address the molecular mechanisms of FAM20C function in RG-like cancer cells or the invasion niche. Does it regulate migration or motility in general or directed migration towards neurons?”*

Response to Reviewers

Response: We used mouse primary neurons because we thought this setting is similar to in vivo xenograft of hNSCs into a mouse brain microenvironment. To address potential cross-species differences in receptor/ligand interactions, we followed the reviewer's suggestion to use human neurons differentiated from human NSCs. Briefly, we differentiated hNSCs in the NSC differentiation medium for 40 days to allow neuronal differentiation and maturation, as evidenced by neurite growth and expression of neuronal markers TUJ1 and MAP2 (revised Fig. S8D, E, and Methods) (Page 13 and 30). We re-performed transwell migration assay and obtained consistent results (revised Fig. 6E). To test whether FAM20C regulate migration or motility in general, we performed a standard transwell migration assay using serum as chemoattractant, but did not observe any difference in the *FAM20C* KO cell lines (revised Fig. 6D). We also moved our original data using mouse primary neurons to the supplemental figure (revised Fig. S8B, C), and revised the text accordingly (Page 12-13). Together, these data support that FAM20C regulates directed migration of RG-like cells towards neurons.

7) *“The functional significance of many observations is not clear. For example, while detecting alternative splicing patterns of various genes is interesting, it is not clear how this information impacts our understanding of gliomagenesis. This should be more explicitly stated and tested in the proper context.”*

Response: We agree with the reviewer. In our revised Figure 3 and S6, we increased our sequencing depth of our long-read sequencing data, reanalyzed the differentially expressed transcripts and splicing junctions based on the new niche assignment, predicted potential regulatory splicing factors, and analyzed their relationship to patient survival to provide functional insights to these observations. On Page 8-9, we added a new paragraph to discuss these new results as follows: “To investigate the clinical relevance of niche-enriched isoforms, we compared our data with the TCGA GBM transcription dataset. Since the TCGA dataset is based on short-read sequencing which does not offer accurate full-length isoform information, we compared these datasets at

the splicing junction (SJ) level (Methods). We identified 76899 new SJ through long-read sequencing, while the remaining two thirds of SJs detected in our dataset were identified in the TCGA dataset, confirming the fidelity of our long-read sequencing (Fig. 3f). Among the shared SJs, we filter out SJs unique to specific isoforms for perform survival analysis. For example, long (*Tomm6-202*) and short (*Tomm6-201*) isoforms of *Tomm6* (translocase of outer mitochondrial membrane 6) are differentially enriched in the hypoxic niche versus the vascular/invasive niches (Fig. 3g, h). The long isoform *Tomm6-202* contains a unique SJ “chr6:41789337-41789530:+”, whose high expression is correlated with favorable prognosis (Fig. 3i and Extended Data Table 6). To predict the regulatory mechanisms for niche-specific isoforms, we further identified splicing factors (SFs) whose expression patterns are consistent with the short isoform (Methods, Extended Data Table 5). The AS for *Tomm6* is likely regulated by *YBX3*, which is confirmed by eCLIP and shRNA knockdown data from the ENCODE database (Fig. 3j). Consistently, low expression of *YBX3*, corresponding to high expression of the long isoform *Tomm6-202*, is associated with favorable prognosis (Fig. 3j, k). As another examples, we identified the *U2AF2* regulated AS of *SNHG6 201/203*, which has been implicated in the progression of hepatocellular carcinoma, and the association with patient survival (Extended Data Fig. 6 c-f). Thus, our data provide a spatial profiling of diverse RNA isoforms in glioma samples, and identified survival related isoforms/SJs and potential regulatory SFs.”

In the Discussion section, we also made an explicit statement as follows: “Also, the biological significance of differentially expressed isoforms needs further validation from functional studies.” (Page 15)

8)“*Considering that there is a significant overlap in expression patterns of receptor/ligands in different cell types in brain tumors (for example, cancer cells can express some traditionally considered immune cell markers), and since each spot in the spatial-seq data contains multiple cells of unknown numbers and lineages, cell-to-cell communication network analysis shown in Fig 4g is problematic.*”

Response: We agree with the reviewer that it is difficult to assign a specific ligand or receptor to a certain cell type in spatial transcriptomics due to the overlap of gene expression in multiple cell types, although we thought it could at least offer a rough estimation about the cell-to-cell communications. In the revised Fig. 4, we decided to remove them in the figures but kept the ligand-receptor pair prediction in Table S8 without specific cell-type assignment. We also revised the text as follows: “To predict niche-specific cell-to-cell communications networks, we first identified 101 receptor-ligand pairs that are co-expressed in spots within each niche (Methods and Extended Data Table 8). Considering that there is a significant overlap of receptor/ligands expression in different cell types in brain tumors and each spot contains multiple cell types, we cannot definitively assign a receptor or ligand to a specific cell type. Interestingly, 63/101 pairs contain at least one ligand or receptor that are among RG signature genes (Extended Data Table 8). The ratios are highest in the invasive (67%) and hypoxic niches (78%), suggesting that RG-like cells serve as communication centers in these niches.” (Page 10)

9) *“Key references are missing in many places and previous work demonstrating the same conclusion is missing. For example, lines 134-137 state that hypoxic and vascular niches are negatively correlated with prognosis, consistent with the notion that cells within these niches may evade standard therapies... There are many publications in the last decade that demonstrated this point and should be referenced.”*

Response: We thank the reviewer for pointing out our missing references. We have systematically gone through our manuscript to include necessary references. For lines 134-137, since Reviewer 2 raised questions about the implications of survival analyses for these module signatures, we removed the data in revised Fig. 2 and the discussion in the revised manuscript (See also response to Review 2, Minor Concern 11).

Response to Reviewers

10) *“They should show tumor volume “outside” of the brain in control and FAM20C KO gliomas. It looks like the leptomeningeal spread is increased in FAM20C-KO tumors.”*

Response: We have calculated the tumor volume outside of the brain and did not observe statistically significant difference (revised Fig. 6F). Notably, since these tumors are “free-floating” and could be partially lost during tissue preparation, we did observe greater cross-sample variations. In this regard, we believe the Ki67% is more reflective of the state of tumors outside the brain, which are comparable between FAM20C KO and CTR groups (revised Fig. 6H).

Minor concerns:

“1) Statistical testing should be more detailed, and the numbers of replicates should be clearly indicated.”

Response: We provided more detailed description of our statistics, and the original quantification was included as in a “Source Data” file in the resubmission.

Reviewer #2 writes: *“In the presented manuscript, the authors use 10X Visium Spatial Transcriptomics, including Illumine short and Nanopore long read sequencing, to map the spatial distribution of recurrent transcription programs. The term “multiomics” in the title is misused because the authors only examined the transcriptome; splice variants, although highly valued, do not justify the term “multiomics.” The study included 11 data sets (from 10 patients) with 5 DMG, 2 IDH-mutated, and 3 IDH-WT samples. Unfortunately, the cohort of different entities is very small and it is difficult to draw conclusions from the small sample size. My greatest concern is that the authors mixed the entities and did not analyze each entity separately. IDH-mutated and WT tumors are different cancer types as is DMG. The spatial transcriptomics technology is new and the data sets are highly appreciated. Since the authors obviously followed the preprint published last year (Ravi et al.) (Similar structure of figures and analytical*

Response to Reviewers

approach), they should discuss and compare their results with the five spatially separated programs presented previously. Including and comparing their results with a large cohort may help to support their findings. The manuscript is clearly written, but sometimes lacks detail; in particular, I was disappointed by the superficial description of the methods. The authors need to work on the methods section to ensure reproducibility.”

Response: We thank this reviewer for his positive comments and frank suggestions. Following his suggestions, we have 1) replaced “multi-omics” with “transcriptomics” throughout the manuscript; 2) analyzed different tumor types separately wherever possible; 3) discussed and compared our results with Ravi et al.; and 4) rewrote the Methods section to provide more details.

Specifically, we compared our results with Ravi et al. both in the results and discussion. We added a panel in revised Fig. 2e and a paragraph in the results section as follows: “Ravi. et al. recently published a spatial transcriptomic dataset of 20 GBM patient samples, revealing five recurrent spatial transcriptional programs. To compare our results with a larger cohort, we performed spatially weighted correlation between our modules and their transcriptional programs in our dataset (Fig. 2e). Our “Hypoxic niche” strongly correlates with the “Reactive Hypoxia” program. “Tumor core” correlates with “Spatial OPC” and “Neuronal development”, while “Invasive niche” correlates best with “Radial glia”. “Vascular niche” does not appear to have a clear counterpart, exhibiting correlation with “Radial glia”, “Reactive hypoxia”, and “Reactive immune”. Thus, our analysis identified similar but different spatial transcriptomic programs, possibly due to different spot filtering and data processing methods.” (Page 6-7).

In the discussion section, we also added a paragraph “Notably, our niche-specific transcriptional programs correlate with but differ from Ravi. et al.’s spatial transcriptional programs. In their study, they used a very stringent threshold (90% based on their tumor content estimation) to filter malignant spots, trying to identify tumor cell specific transcriptional programs in different microenvironment. However, our

Response to Reviewers

deconvolution analysis shows that spots rarely contain more than 50% of tumor cells even in the tumor core. Consistent with single-cell GBM datasets, there are a significant portion of immune and vascular cells within the tumor. The current 10X Visium platform lacks single-cell resolution and each cell contain multiple cells from different cell types. The diffusive nature of malignant gliomas also argues against the prevalence of spots with >90% tumor cells. Thus, we believe it is more feasible to consider each spot as a multicellular ecosystem, and identify recurrent niche-specific signatures correlated with histopathology.” (Page 14-15)

“Another high-level problem is that the authors did not use the latest computational tools developed for spatial transcriptomics while instead using tools tailored for transcript space analysis but ignoring the level of cartesian space. One example is the use of correlation: here the authors used a correlation analysis (Pearson? Not mentioned!) that considers each spot as independent, which is not correct in spatial analysis. Since adjacent spots are not independent, the authors must use correct statistical models such as GWR. In other words, the correlation must be corrected for spatial dependencies.”

Response: We thank this reviewer for pointing out problems with our statistic model. We have tested multiple latest computational tools including SpotClean, Seurat, SpatialPCA, Spruce, SpatialDE2, BayesSpace, Banksy, GWR package, and SPATA2, reanalyzed our data, and thoroughly revised our figures. In the revised analytic pipeline (revised Fig. 1A and Methods), we used SpotClean (<https://doi.org/10.1038/s41467-022-30587-y>) to adjust for contamination from adjacent spots (“spot swapping”), BANSKY (<https://doi.org/10.1101/2022.04.14.488259>) to perform spatial weighted clustering and signature gene identification (Page 4 and 21), and calculated spatial weighted correlation between cluster signatures using GWR (Page 5 and 23) (See also Response to Reviewer 2, Minor Concern 2 and 6).

Response to Reviewers

“The authors used pseudotime estimation to study expression differences localized in cartesian space. First, pseudotime estimation is flawed (mathematically and biologically) in non-single-cell, spatially resolved transcriptomics unless the authors can correct the model based on the cellular composition of individual spots. Why don't the authors use spatial trajectory analysis to estimate gene expression gradients in space (SPATA2 toolbox?)”.

Response: We thank the reviewer for the suggestion and have used *SPATA2* to perform spatial trajectory analysis in revised Fig. 5D and identified the GCL_TI signature based on *SPATA2*-predicted dynamic gene expression pattern (Page 11).

Minor concerns:

1) *“The authors do not mention that 10X Visium does not have single cell resolution which complicates the analysis and interpretation of the data. First, they should show how many cells are located within each spot and the diversity across samples. Using all spots for cluster analysis will lead to high confounder based on spots which are only contain a low frequency of cancer cells. CNAs are a nice marker for GBM to identify the tumor content within each spot, properly the K27 mutation (or IDH mutation) can be used. At least the authors need to discuss and address this problem.”*

Response: Following the reviewer's suggestion, we have used CNV to infer the tumor content in each spot since all our tumor samples exhibit significant CNVs (revised Fig.1C, D) (Page 4), and added a new section of tumor content estimation in Methods (Page 20). We also tried to use *H3K27M* for DMG. While the majority of spots with *H3K27M* mutation also exhibit broad CNVs, the low detection rate (~1 per 100,000 reads) makes it less reliable to estimate the tumor content. Similarly, we did not detect sufficient IDH mutant reads to perform tumor content estimation in *IDH*-mut GBM samples. Thus, we used CNV-based tumor content estimation to filter out malignant spots for subsequent analyses.

Response to Reviewers

2) “*Unsupervised Non-negative matrix factorization (NMF) was performed on all malignant spots using the NMF R package²³ (version 0.23.0). This approach is highly biased for several reasons: 1. As the authors show in Figure 1, the data are clustered by patient, which is not surprising due to the lack of horizontal integration in the analysis!!! However, even after horizontal integration, spatially resolved transcriptomics and single-cell data (not as strong as spatial data) of cancers show high heterogeneity, called inter-patient heterogeneity, which has been recently discussed in several papers (Neftel et al., 2019). Therefore, the community strongly recommends analyzing patients individually and then comparing clusters to identify metamodules. For spatial data, authors should also compare the spatially weighted correlation of metamodules for clustering (not just correlation as shown here).*”

Response: Following the reviewer’s suggestions, we have used BANSKY, a new spatial clustering algorithm that incorporate Cartesian space to analyze patient samples individually (revised Fig. S1B and S2B), identify signature genes for each cluster from different patients, and then horizontally integrated cluster signatures to identify metamodules (revised main text and Methods, Page 5 and 22-23). We also calculated spatial weighted correlation between cluster signatures using GWR (Revised Fig. 2B).

3) “*In Figure 2 C, the spots are differently visualized (compare hypoxic vs. the rest (spot and spot distance))?*”

Response: Since the original tissue sections from each patient varies in size, Seurat has a default “auto-scale” setting to visualize each sample at a different magnification. We tried to change the setting but found the resulting images were visually compromised, particular for those small sections. Thus, we chose to keep the default setting while labeling the magnification for each image such as 2.3X, 3.5X to reflect the auto-scale in revised Fig. 2D, Fig. S1B, and Fig. S2B.

4) *“The authors state that the hypoxic signature is enriched in necrosis. Shouldn't necrosis be excluded from cluster analysis? Necrosis can be well visualized by the total number of UMIs. What was the number of UMIs in the areas defined as necrosis? Do the hypoxic niches show a lower number of UMIs compared with the other groups? Wouldn't this be a confounding factor based on the fact that higher stress sample preparation is automatically defined as hypoxia?”*

Response: We compared the gene number and mitochondrial gene % in spots from hypoxic niches with spots from other niches across all glioma samples, and did not observe dramatic differences (revised Fig. S5A). We also checked two DMG samples with typical Pseudopalisading necrosis histology, and did not observe dramatic differences in the hypoxic niches (revised Fig. S5B, C). Based on H&E staining, pseudopalisading necrotic areas do contain cells with prominent nuclear staining. Thus, we think our standard QC has already removed low quality spots and the remaining spots in the hypoxic niches can be used for subsequent analyses. We also discussed these results in the main text: “To test whether the spatial transcriptional modules are influenced by low-quality spots (particularly those in the hypoxic/necrotic area), we compared the gene number and mitochondrial gene percentage in spots from hypoxic niches with spots from other niches across all glioma samples, and did not observe dramatic differences based on the median values.” (Page 6)

5) *“Spatial data have hidden dependencies (as mentioned above). Another level of potential confounder is the spatial distribution of genetic heterogeneity or subclonal architecture (which is spatially segregated). Authors should investigate whether their transcriptional programs are not derived from subclonal architecture. Since subclones also have a specific spatial distribution, the question arises whether transcriptional programs follow the subclonal architecture or occur independently. (See Neftel et al. subgroups vs subclones...). Since the Verhaak group has shown that metabolic (hypoxia) stress leads to an increase in CNAs and epigenetic dysregulation (Johnson KC et al, 2021)”*

Response: Following the reviewer's suggestion, we identified subclones in each sample based on tumor signature CNV events (total 24 subclones), and analyzed their spatial distribution across glioma niches. We found 10/24 niche-dominant subclones (defined as more than 75% of spots per subclone in a niche), and 14/24 non-niche dominant subclones (Fig. 2f, Extended Data Fig. 3b, and 4b). Thus, similar to the Ravi et al. study, subclonal architecture does not appear to play a major role in specifying the transcriptional programs. We described these results in the main text (Page 7).

6) *“Then the original Louvain algorithm (FindClusters) with clustering resolution 0.8 or 3.0 was performed to cluster spots and we manually combined clusters based on histology... A good example of the lack of detail in the method. 1. Why 0.8 or 3.0 sounds very high to me. Normally an iterative approach should be taken, testing all resolutions from 0.1 to, say, 3. Then the stability of the cluster should be tested for all resolutions to find the most stable resolution. For spatial data, clustering methods that incorporate Cartesian space, such as BayesSpace, should be used in the first instance.”*

Response: Following the reviewer's suggestion, we tested several recently developed spatially aware tools such as Seurat, BayesSpace, SpatialPCA, Spruce, SpatialDE2, and Banksy (Methods). Since the DMG1 sample contains a significant portion of normal cerebellum tissue with clearly demarcated anatomic domains, we used DMG1 as a benchmark to compare the clustering results, and found that the clusters generated by Banksy best correlate with anatomical domains in DMG1. Thus, we performed BANSKY on spots from each sample, generating unique spatial clusters that can be mapped onto distinct histopathological regions. We added this discussion in the revised main text and Methods on Page 4 and 21.

7) *“The additional data on splice variants is of great interest, but needs to be better explored. How different is the situation in the subgroups. Can the authors find spatial gradients of splice variants within the subgroups?”*

Response: Since long-read sequencing is more error prone compared to short-read sequencing, the barcode recovery rate for long-read spatial transcriptomics is not very high in our original dataset, making it difficult to identify spatial gradients of isoforms in tumor subgroups. To address this question, we further increased the sequencing depth of our long-read sequencing data (adding 80 gigabytes data), reanalyzed the expression of transcript isoforms and splicing junctions, and identified spatially differentially expressed isoforms that exhibit recurrent patterns across glioma samples (pan-glioma) or within subgroups (DMG, GBM^{IDHwt}, or GBM^{IDHmut}), summarized in revised Table S5 and described on Page 7. Still, we believe the sample sizes for GBM subtypes are too small to push a strong claim. Instead, we predicted potential regulatory splicing factors, and analyzed their relationship to patient survival to provide functional insights to these observations (See also Response to Reviewer 1, Major Concern 7).

8) *“Deconvolution with 3 different data sets is very difficult to interpret. The authors should start integrating the datasets (horizontal integration / or azimuth mapping), including a cortex dataset (for the GBM). No statistics will be provided!”*

Response: Following the reviewer’s suggestion, we integrated Bhaduri et al. GBM scRNA-seq, Nowakowski et al. human cortex scRNA-seq, Filbin et al. DMG scRNA-seq, and Aldinger et al. human cerebellum snRNA-seq datasets as reference datasets to perform deconvolution analysis (revised Fig. 4A and Methods) (Page 9 and 23-24). We also provided the statistics for cell content comparisons (revised Fig. 4B, C).

9) *“The discussion between AC and RG is not clear to me. The authors also need to include the Nefel classification (as a standard single cell classification). I fully agree with the authors that the RG population is a distinct cell population responsible for migration and invasion. They should show that these cells are described as part of the AC-like subset in DMG and GBM. The statement that this population has not been described is not true, it has just been named differently.”*

Response to Reviewers

Response: We reanalyzed the Neftel et al. GBM dataset, and found that RG scores are highest in a subset of cells at the AC-like and MES-like states. Since the MES-like cell state is associated with glioma invasion but not a specific cell-type, it is not surprising that RG-like cells exhibit MES-like state. Thus, these analyses support that RG-like cells are present in both DMG and GBM, which were classified as AC-like or MES-like cells in previous studies (revised Fig. 4G) (Page 10), and toned down our statement to say that this population has not been classified as RG in the previous studies, instead of “has not been described” (Page 10).

10) *“The analysis performed in Figure 5 is interesting, but the method used is partially flawed (as mentioned above). The authors should include spatial trajectories and spatially weighted correlation analyses.”*

Response: As mentioned above, we used *SPATA2* to perform spatial trajectory analysis in revised Fig. 5D.

11) *“The authors state that their scores based on TCGA analysis have clinical implications. Do the authors not contradict themselves by showing that their signatures are recurrent in all patients and then showing that the signatures are associated with better or worse survival? Is the percentage of signatures relevant? Does it matter how much of which subgroup was resected? There is so much room for error here that I don't think the comparison is useful.”*

Response: We agree with the reviewer and have removed the correlation analyses of niche module score expression with TCGA patient survival in revised Fig. 2.

12) *“last but not least, I would like to note that the authors need provide a QC overview with the most important parameters such as Mean UMI Max UMI and Min UMI nr. genes per spot (man min), sequencing saturation percentage of reads in sample and per spot (min max) and so on ...”*

Response to Reviewers

Response: We added a new supplementary table (Table S2) to present the complete QC information for each sample in our short-read and long-read datasets.

13) *“Finally, I would like to motivate the authors, the data set warrants publication in a reputable journal. I am aware that it is not easy to present the data and consider all levels of complexity, but the authors had better try. I would like to motivate the authors to understand my concerns, because honesty helps to optimize the mouse script!!!! I am looking forward to the revised version”*

Response: We'd like to thank this reviewer for these encouraging words, and his/her suggestions have clearly improved our manuscript.

Reviewers' Comments:

Reviewer #1:

Remarks to the Author:

This is a much-improved manuscript but still fraught with problems. Authors tried to address the concerns expressed by both reviewers, but many responses are unsatisfactory and have not addressed the critical, inherent problem in the experimental design. As the other reviewer pointed out, DMG and GBM are very different tumors and should not be analyzed together, especially when assessing microenvironment/niche differences since they are strongly influenced by metabolic differences. The metabolism between IDH wt and mutant gliomas are well documented. The authors may have been forced to perform their analysis with combined samples due to the small sample size from each tumor type (inherent experimental design flaw). While the text has been toned down and data reanalyzed with proper methods in response to both reviewer comments, there are still unsubstantiated claims and general lack of rigor in analysis, interpretation, and data presentation. Finally, there are very little technical or conceptual advances that are strongly supported by the data presented. The most interesting and novel analysis is the slice variant analysis but it seems to be from a n=1 analysis.

- 1) It is not convincing that FAM20C is essential for RG-like cells in a neuron-rich microenvironment. What is the evidence that the artificial DMG-like cells they used are RG-like? Also, if the IC injections were performed properly in the pons, then the KD cells had to migrate through the normal brain to reach the leptomeningeal space where they grow out "outside the brain". So how can one conclude that FAM20C is required for invasive growth of RG-like cells in a neuron-rich environment when the KD cells could invade through the brain parenchyma?
- 2) Related to above point, they show that proliferation of FAM20C-KD cells is reduced in vivo. In this case, they need to control for differences in proliferation in the in vitro invasion assay.
- 3) Measuring cresyl violet in invasion assay to conclude that KD cells do not migrate towards human neurons is not convincing. There is no way to distinguish the dye content from neurons and invading glioma cells as the assay is performed.
- 4) The title and data presented imply that FAM20C regulates radial glial cancer stem cell-like cells but this is also not supported by the data: clonogenicity is not significantly different in FAM20C_KD F1 cells compared to control. While colony formation is not a commonly used method to assess CSC-like cells in gliomas, it is the closest data they have presented to testing the CSC-like phenotype. Overall, there are too many inconsistencies with their observations with FAM20C-KD cells.
- 5) Splice variant analysis does not address the possibility that alternative splice variants represent the presence of different cell types in different niches. Given that it is n=1 analysis, and it does not address whether the variation arises from different cellular composition or microenvironmental influence, it is not clear what solid conclusions can be drawn from this study.
- 6) Cell:cell communication analysis is again not performed rigorously and does not support the statement that "RG-like cells serve as communication centers" in the invasive and hypoxic niches.
- 7) "dramatic" vs "significant" differences should be supported by statistical analysis. Just one example (among many) line 169-173 referring to Supp Fig 5a-c may not be "dramatic" but may be statistically significant.
- 8) Line 164-165 is questionable. DMG4 invasive and hypoxic module scores are not mutually exclusive? Supp Fig 3a. Need to show high magnification image of the area?
- 9) Line 175-177. Was the comparison performed with only the GBM (IDH WT) modules or all samples? A fair comparison would be only with matching types of tumors, and it is not clear what samples were compared here.
- 10) How generalizable are the different slice variants in different niches? Data presented appear to come from only one sample. They should demonstrate that the same pattern holds up in other samples. Otherwise, what is the significance? Especially since it is not even clear whether the pattern arises from different cellular compositions in different niches (in one sample).
- 11) The observation with Tom6-202 variant is confusing. It is well established that hypoxia is associated with worse prognosis in gliomas but the Tom6-202 unique slice variant enriched in the

hypoxic niche is associated with favorable prognosis (lines 231-232)?

12) Heatmaps shown in figure 3 do not seem to match the actual reads indicated on genomic traces. How were they normalized to generate the heat map?

13) Lines 254-255 should be re-written. As is, it's confusing and misleading. Fig 4a shows AC-like cells are most abundant in the invasive niche and not "the invasive niche has the highest neuronal content (Fig 4a, b)". It should be "neuron content is the highest in the invasive niche (Fig 4b)"? Minimally, Fig 4a and Fig4b should be called out separately to clarify the meaning.

Minor concerns

Line numbers and page numbers in the rebuttal letter do not match the revised manuscript in places.

Figure call outs are incorrect in some place- such as liens 357 & 359 on page 13.

Reviewer #2:

Remarks to the Author:

The authors have cleared up the errors and doubts that were raised and I would like to congratulate them on a successful manuscript. I am looking forward to seeing the manuscript in press.

Response to Reviewers (NCOMMS-22-06690B)

We'd like to thank both reviewers for their comments, and we are glad that Reviewer #2 thinks we have addressed all his/her concerns. Below, we provide a point-by-point rebuttal to address the remaining concerns from Reviewer #1. Main text revisions are highlighted by the yellow color.

Reviewer #1 writes: *“This is a much-improved manuscript but still fraught with problems. Authors tried to address the concerns expressed by both reviewers, but many responses are unsatisfactory and have not addressed the critical, inherent problem in the experimental design. As the other reviewer pointed out, DMG and GBM are very different tumors and should not be analyzed together, especially when assessing microenvironment/niche differences since they are strongly influenced by metabolic differences. The metabolism between IDH wt and mutant gliomas are well documented. The authors may have been forced to perform their analysis with combined samples due to the small sample size from each tumor type (inherent experimental design flaw). While the text has been toned down and data reanalyzed with proper methods in response to both reviewer comments, there are still unsubstantiated claims and general lack of rigor in analysis, interpretation, and data presentation. Finally, there are very little technical or conceptual advances that are strongly supported by the data presented. The most interesting and novel analysis is the slice variant analysis but it seems to be from a n=1 analysis.”*

Response: We thank this reviewer for saying that our manuscript is “much improved”. We appreciate the reviewer's feedback and first we'd like to clarify about our experimental design. In our last revision, as Reviewer #2 suggested, we already analyzed DMG, IDH^{wt} and IDH^{mut} GBMs as different tumor entities in addition to pan-glioma analyses, and showed that our major conclusions are consistent across different glioma types, which was appreciated by Reviewer #2. In this revision, we added an

additional supplementary figure and text to make this point more clear (**Revised Fig. S3, Page 5 and 7**).

We respectively disagree with Reviewer 1's assessment regarding the novelty or technical/conceptual advance of our manuscript, particularly in light of the *Nature Communications* policy that papers published during manuscript revision (since March, 2022) do not compromise our novelty.

Finally, our isoform analysis is not an n=1 analysis, and we apologize if our data presentation led to this confusion. Indeed, these analyses were based on recurrent patterns across glioma samples/niches. Specifically, in our last revision, we first performed sample-wise paired comparisons among the four niches to find niche-specific differentially expressed isoforms (DEIs), and then only kept DEIs that exhibited conserved niche-enrichment across all samples (pan-glioma) or specific glioma types (DMG, IDH^{wt} GBM, IDH^{mut} GBM or pan-GBM) (**Table S5**). In this revision, we further modified the figures to show the isoform expression pattern in all samples or specific glioma types, and added more details in the methods to address Reviewer 1's concerns (**Revised Fig. 3c, d, g, h and Fig. S7c, d, Page 26**) (See also Response to Reviewer 1, Major Concern 5 and 10).

Major concerns:

1) *"It is not convincing that FAM20C is essential for RG-like cells in a neuron-rich microenvironment. What is the evidence that the artificial DMG-like cells they used are RG-like? Also, if the IC injections were performed properly in the pons, then the KD cells had to migrate through the normal brain to reach the leptomeningeal space where they grow out "outside the brain". So how can one conclude that FAM20C is required for invasive growth of RG-like cells in a neuron-rich environment when the KD cells could invade through the brain parenchyma?"*

Response: We thank the reviewer for pointing out this alternative interpretation for our data. We added bulk RNA-seq and RG marker immunostaining of HPT cells to show these cells most closely resemble RG-like cells (**Revised Fig. S8f, g, Page 13**). To test whether the tumors outside the brain are results of continuous growth/invasion of endstage tumors inside the brain, we examined brain sections from three mice 21 days after HPT cell xenograft (**Revised Fig. S9h, Page 15, and the figure below**). We consider this as an early stage since it takes 2-3 months for these cells to develop into tumors. At this stage, we can already identify clusters of mCherry⁺ cells in the 4th ventricle and the subarachnoid space (yellow dashed lines). In the meantime, we only observed diffusely infiltrating mCherry⁺ cells in the pons (white dashed lines), which are not yet full-blown tumors and do not exhibit obvious connections with cells outside the brain. These data indicate that from the early stage on tumors inside and outside are growing independently under different microenvironmental context, which may or may not merge together at the endstage. Furthermore, we analyzed multiple sections from endstage *FAM20C KD* mice and did not observe migration paths of mCherry⁺ cells through the brain parenchyma. Thus, at least in our models, the fact that *FAM20C KD* cells can form tumor masses outside the brain does not mean they have to invade through the brain parenchyma.

To check whether this phenomenon is unique to our models, we checked up two other published hNSC-derived DIPG/DMG models using similar protocols (**Funato et al., *Science*, 2014; Haag et al., *Cancer Cell*, 2021**). Based on their descriptions and histology, they also noted tumor development outside the brain without prominent tumor growth in the pons (injection site) (**see the above figure**). One possible

explanation is that for mutant hNSC-xenograft DMG models, since there is injury along the injection path from the lambda suture deep into the pons (with possible involvement of the 4th Ventricle), mutant hNSCs could spread along the injection path, flow with the cerebrospinal fluid, and eventually develop into tumor masses outside the brain.

Taken together, we think it is appropriate to treat tumors inside and outside as different entities under distinct microenvironmental context. The observation FAM20C KD cells do not grow well inside the brain parenchyma supports our conclusion that FAM20C is required for invasive growth of RG-like cells in a neuron-rich environment.

2) *“Related to above point, they show that proliferation of FAM20C-KD cells is reduced in vivo. In this case, they need to control for differences in proliferation in the in vitro invasion assay.”*

Response: We did not control for the proliferation differences in the in vitro invasion assays since HPT hNSCs in the upper insert were placed in growth factor deprived culture medium (hNSC culture medium with B27 but without bFGF) (**Fig. 6d, e and S9c, d, cartoons on the left**), following a published protocol by Michelle Monje group (*Cell*, 2017, Ref. #81). This is an inhibitory condition for cell proliferation, allowing us to directly compare migratory capacities. To ascertain this, we performed additional CCK-8 assay to show that HPT cells cultured under this condition barely increased in number during the experimental interval (48h), using cells cultured in normal hNSC culture medium as positive control (**Revised Fig. S9b, Page 13 and 34**).

3) *“Measuring cresyl violet in invasion assay to conclude that KD cells do not migrate towards human neurons is not convincing. There is no way to distinguish the dye content from neurons and invading glioma cells as the assay is performed.”*

Response: Based on our experimental protocol, we believe we could indeed distinguish the dye content from neurons and invading glioma cells. In our transwell migration

assays, the RG-like cells were cultured in the upper insert with permeable membrane, and neurons were adherently cultured in the lower well. After 48h, the RG-like cells that migrated through to the other side of the membrane were stained by cresyl violet, and the total intensity of the dye was measured. We also did not observe detachment of neurons from the bottom of the lower well during the experiment. Thus, the physical separation ensures that all the migrated cells from the upper insert and stained by cresyl violet are RG-like cells. We have added more details to our Methods section to avoid any confusion (**Page 35**).

4) *“The title and data presented imply that FAM20C regulates radial glial cancer stem cell-like cells but this is also not supported by the data: clonogenicity is not significantly different in FAM20C_KD F1 cells compared to control. While colony formation is not a commonly used method to assess CSC-like cells in gliomas, it is the closest data they have presented to testing the CSC-like phenotype. Overall, there are too many inconsistencies with their observations with FAM20C-KD cells.”*

Response: We’d like to point out that clonogenicity is not the only feature/phenotype of CSC-like cells. Indeed, migration and invasive growth are also features/phenotypes of CSC-like cells, which may be regulated by different regulators depending on the microenvironmental context (recently reviewed by Jeremy N. Rich group in *Cell Stem Cell*). Indeed, our *in vitro* and *in vivo* data consistently show that FAM20C regulates the migration and invasive growth of CSC-like cells in a neuronal context.

5) *“Splice variant analysis does not address the possibility that alternative splice variants represent the presence of different cell types in different niches. Given that it is n=1 analysis, and it does not address whether the variation arises from different cellular composition or microenvironmental influence, it is not clear what solid conclusions can be drawn from this study.”*

Response: We'd like to emphasize this is not an n=1 analysis. In our previous submission we only included differentially expression isoforms that exhibit conserved niche-enrichment across all samples or specific glioma types. While we acknowledge that differential isoform expression may derive from different cellular compositions in our experimental setting, such analysis has been performed on mouse cortex spatial transcriptomic dataset (**Joglekar et al., *Nature Communications*, 2021, Ref. #22**) and bulk primary and recurrent GBM samples (**Aaron A. Diaz group, *Genome Biology*, 2021; Li and Guo, *BMC Cancer*, 2021**), and revealed consistent patterns despite facing the same technical challenge.

6) *“Cell:cell communication analysis is again not performed rigorously and does not support the statement that “RG-like cells serve as communication centers” in the invasive and hypoxic niches.”*

Response: While we believe our ligand-receptor pair analysis offers a closest estimate regarding the cell-communication center, we followed the reviewer's advice to remove this statement in the main text and abstract to be more rigorous (**Page 11**).

7) *““dramatic” vs “significant” differences should be supported by statistical analysis. Just one example (among many) line 169-173 referring to Supp Fig 5a-c may not be “dramatic” but may be statistically significant.”*

Response: We double-checked our figures to make sure statistical significances are indicated by stars (*) or p values in every quantification/comparison. For the example the reviewer pointed out, we revised the text to state that the differences are statistically significant but not dramatic (**Page 7**).

8) *“Line 164-165 is questionable. DMG4 invasive and hypoxic module scores are not mutually exclusive? Supp Fig 3a. Need to show high magnification image of the area?”*

Response: We consulted with the pathologists and confirmed this sample does not contain large areas of invasive/infiltrative or hypoxic regions. Consequently, the differential expression of these module scores are more subtle than other samples. Despite this, our scoring system based on the relative expression of all four modules was able to distinguish small hypoxic regions (arrows) bordering necrosis versus invasive regions (See below, high mag image of the area). Since in some areas we cannot definitely say these two scores are mutually exclusive for this sample, we removed this statement in the revised main text (**Page 6**).

9) *“Line 175-177. Was the comparison performed with only the GBM (IDH WT) modules or all samples? A fair comparison would be only with matching types of tumors, and it is not clear what samples were compared here.”*

Response: The comparison was performed with pan-glioma modules from all samples since we found these modules are conserved across glioma types. To rigorously test this, we followed the review’s suggestion to perform additional comparison using IDH^{wt} GBM modules only, and got consistent results (**Fig. S3b bottom right panel, Page 7**).

10) *“How generalizable are the different slice variants in different niches? Data presented appear to come from only one sample. They should demonstrate that the same pattern holds up in other samples. Otherwise, what is the significance? Especially since it is not even clear whether the pattern arises from different cellular compositions in different niches (in one sample).”*

Response: In this revision, we modified the figures to show the isoform expression pattern in all samples or specific glioma types, and added more details in the methods to address Reviewer 1's concerns (**Revised Fig. 3c, d, g, h and Fig. S7c, d, Page 26**)

11) *“The observation with Tomm6-202 variant is confusing. It is well established that hypoxia is associated with worse prognosis in gliomas but the Tomm6-202 unique slice variant enriched in the hypoxic niche is associated with favorable prognosis (lines 231-232)?”*

Response: Since the two isoforms of Tomm6 are respectively enriched in invasive and hypoxic niche and exhibit isoform switch, higher level of 202 is accompanied by lower level of 201 (enriched in the invasive niche). Then it comes to whether hypoxic niche or invasive niche plays a more dominant role in patient prognosis, which is hard to determine. Thus, we believe for isoform/SJ-survival analysis based on bulk TCGA data, we should describe the result as it is.

12) *“Heatmaps shown in figure 3 do not seem to match the actual reads indicated on genomic traces. How were they normalized to generate the heat map?”*

Response: We added more detailed description how the heatmaps are generated (**Page 27**). Briefly, the mean expression of an isoform in a given niche is calculated as total reads divided by the number of high-quality spots in the niche. The values from four niches were then auto-scaled and visualized in the heatmap to shown enrichment of this isoform in specific niches. To avoid confusion, in this revision we instead labeled the mean expression of each isoform in each niche (**Revised Fig.3c, h and Fig. S7d**).

13) *“Lines 254-255 should be re-written. As is, it's confusing and misleading. Fig 4a shows AC-like cells are most abundant in the invasive niche and not “the invasive niche has the highest neuronal content (Fig 4a, b)”. It should be “neuron content is the*

highest in the invasive niche (Fig 4b)”? Minimally, Fig 4a and Fig4b should be called out separately to clarify the meaning.”

Response: We agree with the reviewer and have rewritten this description as the reviewer suggested (**Page 10**).

Minor concerns

“Line numbers and page numbers in the rebuttal letter do not match the revised manuscript in places.”

Response: We apologize for this confusion, which might be caused by WORD to PDF conversion. In this revision, we ensure the page numbers are properly referenced.

“Figure call outs are incorrect in some place- such as liens 357 & 359 on page 13.”

Response: The figure call outs have been thoroughly checked and revised in this revision.

Reviewer #2 writes: *“The authors have cleared up the errors and doubts that were raised and I would like to congratulate them on a successful manuscript. I am looking forward to seeing the manuscript in press.”*

Response: We’d like to again thank this reviewer for his constructive suggestions.

Reviewers' Comments:

Reviewer #1:

Remarks to the Author:

Thank you for addressing my concerns and providing new data. Most concerns have been addressed; however, the point regarding cancer stem cells remains.

The gold standard definition of a cancer stem cell is a cell that can generate a tumor that replicates the original tumor phenotype upon transplantation. Clonogenicity in vitro and migration/invasive growth phenotypes are both characteristics associated with CSCs but are defining characteristics. Plenty of cancer cells have migratory or clonogenic characteristics without fulfilling the functional CSC phenotype in vivo. As such, responses regarding the role of FAM20C in regulating GSCs and the title of the study are weak.

Response to Reviewers (NCOMMS-22-06690C)

We'd like to again thank both reviewers for their constructive comments. Below, we provide a rebuttal to address the remaining concern from Reviewer #1.

Reviewer #1 writes: *“Thank you for addressing my concerns and providing new data. Most concerns have been addressed; however, the point regarding cancer stem cells remains. The gold standard definition of a cancer stem cell is a cell that can generate a tumor that replicates the original tumor phenotype upon transplantation. Clonogenicity in vitro and migration/invasive growth phenotypes are both characteristics associated with CSCs but are defining characteristics. Plenty of cancer cells have migratory or clonogenic characteristics without fulfilling the functional CSC phenotype in vivo. As such, responses regarding the role of FAM20C in regulating GSCs and the title of the study are weak.”*

Response: We agree with the reviewer and have toned down the cancer stem cell aspect of this study in the revised manuscript. In both the title and main text, we use “radial glia-like cells” or “radial glial stem-like cells” instead of “cancer stem cells” to be more rigorous.